# Evaluation of two hydro-meteorological ensemble strategies for flash flood forecasting over a catchment of the eastern Pyrenees

Hélène Roux[1], Arnau Amengual[2], Romu Romero[2], Ernest Bladé[3] and Marcos Sanz-Ramos[3]

[1]Institut de Mécanique des Fluides de Toulouse (IMFT), Université de Toulouse, CNRS - Toulouse, France
[2]Grup de Meteorologia, Departament de Física, Universitat de les Illes Balears, Palma, Mallorca, Spain
[3]Institut FLUMEN, E.T.S. d'Eng. De Camins, Canals i Ports de Barcelona, Universitat Politècnica de Catalunya, Barcelona, Spain

*Correspondence to*: Hélène Roux (helene.roux@imft.fr)

**Abstract.** This study aims at evaluating the performances of flash flood forecasts issued from deterministic and ensemble meteorological prognostic systems. The hydro-meteorological modeling chain includes the Weather Research and Forecasting model (WRF) forcing the rainfall-runoff model MARINE dedicated to flash flood. Two distinct ensemble prediction systems accounting for (i) perturbed initial and lateral boundary conditions of the meteorological state and (ii) mesoscale model physical parameterizations, have been implemented on the Agly catchment of the Eastern Pyrenees with three sub-catchments exhibiting different rainfall regimes.

Different evaluations of the performance of the hydro-meteorological strategies have been performed: (i) verification of short-range ensemble prediction systems and corresponding stream flow forecasts, for a better understanding of how forecasts behave, (ii) usual measures derived from a contingency table approach, to test an alert threshold exceedance, and (iii) overall evaluation of the hydro-meteorological chain using the Continuous Rank Probability Score, for a general quantification of the ensemble performances.

Results show that the overall discharge forecast is improved by both ensemble strategies with respect to the deterministic forecast. Threshold exceedance detections for flood warning also benefit from large hydro-meteorological ensemble spread. There are no substantial differences between both ensemble strategies on these test cases in terms both of the issuance of flood warnings and the overall performances, suggesting that both sources of external-scale uncertainty are important to take into account.

## 1    Introduction

Flash floods are among the most devastating natural hazards worldwide, producing important human and socio-economic losses. The Western Mediterranean region is annually affected by several extreme precipitation events which lead to flash flooding. During the extended warm season, the early intrusion of upper-level cold air masses and the relatively high sea surface temperature boost the convective available potential energy of the low-level Mediterranean warm and moist air.

This natural hazard results from the persistence of deep moist convection and intense precipitation over specific hydrographic catchments during several hours. As many Western Mediterranean small-to-medium sized river basins are highly urbanized, steep and close to the coastline, their hydrological responses are inherently short. Large, rapid and unexpected flows exacerbate flood damage. The development and evaluation of the state-of-the-art hydro-meteorological forecasting tools is a major issue in the Hydrological Cycle in the Mediterranean Experiment (HyMeX; Drobinski et al. 2014). This program aims at addressing the following science questions, amongst others: How can we improve heavy rainfall process knowledge and prediction? How can we improve hydrological prediction?

Hydro-meteorological forecasting tools can contribute to a better understanding and forecasting of flash floods so as to implement more reliable forecasting and warning systems over the Western Mediterranean. Short-range quantitative precipitation forecasts (QPFs) by high-resolution numerical weather prediction (NWP) models are an effective tool to further extend flood forecasting lead-times beyond the basin response times. NWP models capture the initiation and evolution of small-scale and convectively-driven precipitations, with similar spatial and temporal scales to the flash flood-prone catchments (Leoncini et al., 2013; Fiori et al., 2014; Ravazanni et al. 2016; Amengual et al. 2017). Although QPFs can be directly used to force one-way hydrological models, the hydro-meteorological forecasts are impacted by different types of uncertainties. Uncertainties are inherent to each of the hydro-meteorological chain components: model parameterization and structure, limitations of measuring devices providing observation data, initial and lateral boundary conditions (Zappa et al., 2010).

External-scale inaccuracies to the hydrological models emerge from two distinct sources when forecasting deep moist convection and heavy rainfall with NWP models. First, errors arise from the complexity and nonlinearity of the physical parameterizations. Second, uncertainties emerge when representing the exact initial atmospheric state and boundary forcing across the scales where convection develops. But reliable spatial and temporal QPF distributions are necessary to render skilful quantitative discharge forecasts when coping with floods over small and medium size basins. Otherwise, the issuance of precise and dependable early flood warnings is inhibited (Le Lay and Saulnier, 2007; Bartholmes et al., 2009; Cloke et al., 2013).

To alleviate the impact of these external-scale uncertainties, short-range ensemble prediction systems (SREPSs) are used to build hydrological ensemble prediction systems (HEPSs). SREPSs aim at sampling the set of plausible outcomes and at accounting for the most relevant uncertainties in the atmospheric forecasting process so as to increase. Uncertainties in the initial and boundary fields can be encompassed by conveniently perturbing initial and lateral boundary conditions (IC/LBCs, Grimit and Mass, 2007; Hsiao et al., 2013). Uncertainties in model parameterizations are coped by populating the ensemble with multiple combinations of equally-skilful physical schemes (Stensrud et al., 2000; Jankov et al., 2005; Amengual et al., 2008; Tapiador et al., 2012; Amengual et al., 2017). The inclusion of these uncertainties aims at improving the skill and spread of the HEPSs by introducing independent information of all the plausible atmospheric states and processes. Therefore, SREPSs are increasingly used in hydrologic prediction (Cloke and Pappenberger, 2009; Verkade et al., 2013; Verkade et al., 2017; Siddique and Mejia, 2017; Benninga et al., 2017; Bellier et al., 2017; Edouard et al., 2018; Jain et al,

2018; Bellier et al., 2018). Several studies have stated that probabilistic forecasts could improve decision-making if appropriately handled (e.g. Krzysztofowicz, 2001; Todini, 2004; Ramos et al., 2013; Antonetti et al, 2019). As stated by Zappa et al. (2011), each member of a meteorological ensemble can be fed into a hydrological model to generate a hydrological forecast.

However, the most appropriate methods for generating HEPSs and the quantification of their added value are still under assessment (Cloke and Pappenberger, 2009; Cloke et al., 2013). Further efforts devoted to examine the predictive skill of both ensemble strategies and how the external-scale uncertainties spread into the HEPSs become paramount for the optimal design of hydro-meteorological operational chains over the flood-prone Western Mediterranean area. The objective of the present work is to evaluate the predictive skill of two distinct HEPS generation strategies –accounting for perturbed IC/LBCs (PILB) and mixed-physics (MPS)– for three flash flood episodes over the Agly basin (Fig. 1). This catchment of the Eastern Pyrenees has been selected as an experimental area as several subcatchments exhibit different rainfall regimes. Given the small size of the subcatchments (from 150 km² to 300 km²), the localization of the precipitation patterns is crucial (Rossa et al., 2010) and it's a challenging to implement QPFs for such small subcatchments. QPFs are generated by using the Weather Research and Forecasting model (WRF; Skamarock et al., 2008). Next, 48-h WRF forecasts are propagated down through the MARINE hydrological model (Roux et al., 2011) to investigate the quantitative discharge forecasts in timing and magnitude of these flash floods. The resulting HEPSs are examined using different criteria to illustrate the potential benefits of the probabilistic hydro-meteorological forecast chains. The rest of the paper is structured as follows: section 2 presents a short overview of the flash floods, the study area and the observational networks; sections 3 and 4 provide an insight into the hydrological and atmospheric models and the strategies for ensemble generation; results are presented in section 5. The last section summarizes main conclusions and provides further remarks.

## 2    Data and case studies

### 2.1    Overview of the Agly catchment

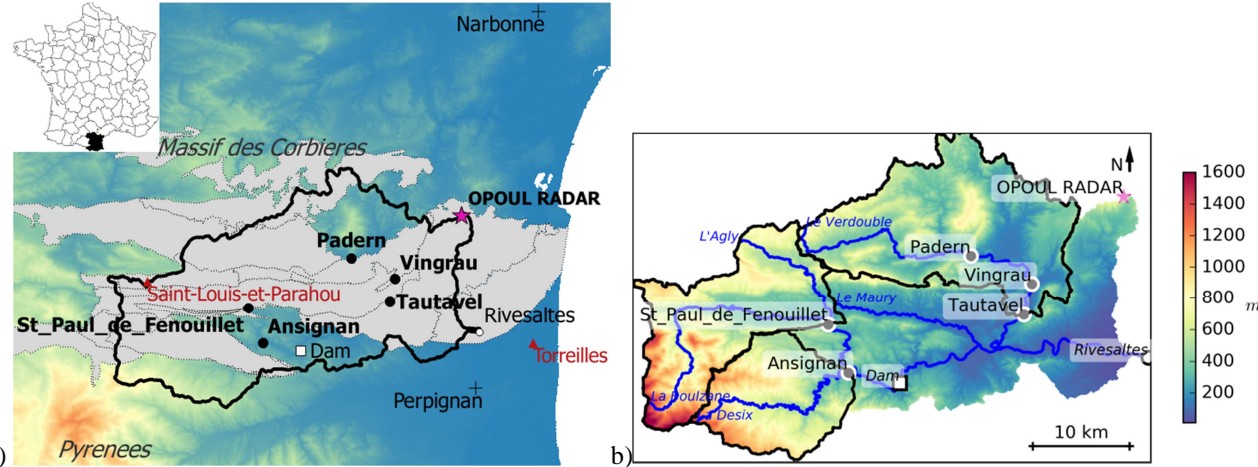

**Figure 1: a) Location of the Agly catchment and of the meteorological radar (grey area: karstic areas underlying the Agly catchment, from BDLISA v.2: Base de Donnée des Limites des Systèmes Aquifères, https://bdlisa.eaufrance.fr/ accessed June 18, 2019). b) Digital terrain model of the Agly catchment (Source: IGN; MNT BDALTI). Also included the main tributaries (blue lines, source: IGN, BD CARTHAGE), the radar location (pink star: OPOUL RADAR), the discharge gaging stations (black dots), the dam (white square) and the outlet (white circle).**

This study focuses on a catchment in the north side of the Eastern Pyrenees, the Agly, as a test site for implementing the HEPS strategies. The Agly is a coastal river in the north side of the Eastern Pyrenees (Figure 1). It originates from an elevation of approximately 700 m and drains the Pyrenees foothills. It flows into the Mediterranean Sea at Barcarès and has a length of around 80 km. A dam dedicated to flood and water management controls approximately 400 km$^2$ the catchment (Agence de l'eau Rhône Méditerranée & Corse, 2012). It is located just downstream of the confluence between the Agly and one of its main right-hand tributaries, the Désix River, draining an area of around 150 km$^2$ (Figure 1). The main left-hand tributary, the Verdouble River drains an area of 300 km$^2$ located in a region of mid-mountains, culminating between 400 and 500 meters of altitude (Figure 1). Granite and gneiss cover about 300 km$^2$ of the mountainous part of the Agly catchment promoting runoff already facilitated by the steep slopes. North of the catchment, the Corbières massif is dominated by limestones forming karstic networks. According to hydrogeological studies of the area, there are only losses in the Agly and Verdouble catchments due to the karstic system. These losses contribute to the streamflow of two resurgences draining the Corbières massif but located outside of the Agly catchment (Font-Estramar and Font-Dame resurgences) (Salvayre, 1989). The average loss rates are estimated between 0.3 and 1.5 m$^3$/s for the Agly depending on the river discharge and between 0.7-2 m$^3$/s on the Verdouble (Ladouche et al., 2004). These are only average estimates based on observed discharges and assumptions about the functioning of the karst system but they can be considered small enough not to be explicitly represented in flash flood simulations. 80% of the catchment is covered by natural vegetation –forest (45%), shrubby vegetation (17%), maquis and scrubland (16%)–, while 18% is used for agriculture, mainly vineyards.

The Agly catchment is subject to different climate regimes in connection with the distances from the sea and the mountainous reliefs: temperate oceanic in the north-west valley, mountain in the south-west part and, Mediterranean downstream. The rainfall regime varies from east to west with increasing annual cumulated precipitations: the mean annual

cumulated precipitations (1965-1996) range from 600 mm at Torreilles (East, Figure 1) up to 1174 mm at Saint-Louis-et-Parahou (West, Figure 1) (DIREN Languedoc-Roussillon/SIEE-GINGER, 2008). Generally, the rainfall regime is highly variable with very intense precipitation events in fall, winter and spring and very dry summers.

## 2.2 Available data

The precipitation measurements available on the Agly catchment come from two different observational networks:

- PLU: The operational hourly rain-gauge network for flood monitoring purposes and data provided by the regional flood forecasting service, the Service de Prévision des Crues Méditerranée Ouest (SPCMO).
- JP1: 1 km$^2$ quantitative hourly precipitation estimates ANTILOPE J+1 (ANalyse par spaTIaLisation hOraire des PrEcipitations) that come from a merging of radar data and rain-gauges measurements (Laurantin, 2008;
Champeaux et al., 2009).

The hydrometric data were derived from the network of operational measurements at variable time steps (HydroFrance databank, http://www.hydro.eaufrance.fr/). The stream-gauges are located in 5 upstream stations not influenced by the dam (Table 1 and Figure 1). Table 2 summarizes the main hydrological features of the 5 stations. This study will focus on 3 recent events started on 04 March 2013, 16 November 2013 and 28 November 2014, being highly variable (Table 3), with
rainfall lasting respectively 3 days for the spring event and 4 days for the 2 fall events. The selected events have been labelled with the start date and the duration as follows: 20130304_3d, 20131116_4d and 20141128_4d. All the floods feature moderate specific peak discharges for flash-flood, highlighting the high infiltration rates. The runoff coefficient is always higher for the eastern part (station n°5, Table 3) than for the western part. The runoff coefficient is even higher than 1 for 20130304_3d at station n°5. There is no definitive explanation for that, but several possibilities can be considered: (i) the
very high soil moisture at the beginning of the event (65%, Table 3) which can contribute to the runoff at the outlet via subsurface flows; (ii) an amount of snowmelt as there was a snowfall episode at the very end of February 2013 over the Eastern Pyrenees and Corbières, with snow above 700 to 800 m; (iii) the uncertainties in the discharge and precipitation measurements; (iv) a possible supply from the karstic system (Figure 1) however this possibility is pretty unlikely as hydrological studies conclude to only losses in the Verdouble catchments due to the karstic system (Ladouche et al., 2004).
One event occurred in spring with an averagely moist soil (20130304_3d, Table 3), while the other two occurred in autumn with dry soils after the summery drought. The autumn episodes exhibit very different intensities: the specific peak discharges range from 0.3 to 0.6 $m^3s^{-1}km^{-2}$ for 20131116_4d, and from 1 to 2 $m^3s^{-1}km^{-2}$ for 20141128_4d. Concerning the mean of the maximum rainfall intensity over the catchment, they range from: 8 to 14 $mm.h^{-1}$ according to PLU and from 9 to 11 $mm.h^{-1}$ according to JP1 for 20131116_4d; 19 to 30 $mm.h^{-1}$ according to PLU and from 15 to 25 $mm.h^{-1}$ according to
JP1 for 20141128_4d (Table 3). 20141128_4d is therefore much more intense than 20131116_4d according to both observed forcings even if JP1 forcing presents lower intensities. 20130304_3d is in between both episodes, with specific peak discharges ranging from 0.6 to 1.5 $m^3s^{-1}km^{-2}$, but lower rainfall intensities, ranging from 7 to 11 $mm.h^{-1}$ according to PLU and from 6 to 11 $mm.h^{-1}$ according to JP1. These episodes are representative of the different seasonal rainfall regimes

that lead to floods over the Agly. In spring, floods are mainly originated from stratiform type rainfall with moderate but persistent precipitation rates that can result in substantial accumulations. In autumn, floods are most likely driven by convective type precipitations of shorter duration but high intensity.

| Station | River | Area (km$^2$) | $T_c$ ($h$) |
|---|---|---|---|
| n°1 Ansignan | Désix | 157 | 9 |
| n°2 St-Paul-de-Fenouillet | Agly | 216 | 10 |
| n°3 Padern | Verdouble | 161 | 8 |
| n°4 Vingrau | Verdouble | 301 | 11 |
| n°5 Tautavel | Verdouble | 305 | 12 |
| Rivesaltes | Agly | 1053 | 23 |

5 **Table 1: Characteristics of the 5 subcatchments and the whole catchment. The time of concentration is estimated using Bransby Williams formula (Eq. 3).**

| Station | Period | QIX2 (m$^3$s$^{-1}$) | QMEV (m$^3$s$^{-1}$) | TMEV |
|---|---|---|---|---|
| n°1 | 1994-2018 | 85.0 [57.00;120.0] | 291 | 15/03/2011 |
| n°2 | 1971-2018 | 87.0 [77.00;99.00] | 483 | 26/09/1992 |
| n°3 | 2006-2018 | - | 281 | 30/11/2014 |
| n°4 | 2010-2018 | - | 525 | 30/11/2014 |
| n°5 | 1967-2018 | 170.0 [140.0;200.0] | 922 | 13/11/1999 |

**Table 2: Hydrological statistics of the 5 catchments (From HydroFrance databank, http://www.hydro.eaufrance.fr/). QIX2: 2-year return period of maximum instantaneous discharge and confidence interval 95%, QMEV: known maximum instantaneous discharge, TMEV: Date of QMEV.**

| | | PLU | | JP1 | | $Q_p^o$ | $Q_p^o\big|_s$ | $T_p^o$ | $C_r$ | $H_{ini}$ |
|---|---|---|---|---|---|---|---|---|---|---|
| Event | Station | Cumulated P (mm) | Max I (mm/h) | Cumulated P (mm) | Max I (mm/h) | $(m^3 s^{-1})$ | $(m^3 s^{-1} km^{-2})$ | $(dd\ hh{:}mm)$ | | (%) |
| 20130304_3d | n°1 | 186 ± 19 [226] | 7.4 | 167 ± 30 [208] | 6.4 | 137 | 0.87 | 06 06:35 | 0.17 | 48 ± 0 |
| | n°2 | 183 ± 37 [215] | 6.9 | 160 ± 25 [217] | 5.8 | 137 | 0.63 | 06 09:40 | 0.12 | 51 ± 3 |
| | n°5 | 181 ± 28 [218] | 11.2 | 192 ± 26 [294] | 11.4 | 459 | 1.50 | 06 12:24 | 1.07 | 65 ± 1 |
| | Outlet | 179 ± 40 [226] | 8.5 | 178 ± 30 [294] | 8.6 | 970 | 0.92 | - | - | 56 ± 7 |
| 20131116_4d | n°1 | 227 ± 11 [303] | 13.1 | 208 ± 18 [242] | 10.9 | 47 | 0.30 | 18 05:10 | 0.05 | 35 ± 1 |
| | n°2 | 275 ± 26 [303] | 14.1 | 212 ± 24 [269] | 8.8 | 131 | 0.61 | 18 01:58 | 0.05 | 42 ± 4 |
| | n°5 | 181 ± 37 [241] | 8.0 | 183 ± 17 [230] | 10.6 | 109 | 0.36 | 18 06:13 | 0.21 | 55 ± 3 |
| | Outlet | 208 ± 49 [303] | 9.9 | 194 ± 25 [285] | 9.6 | 260 | 0.25 | - | - | 45 ± 8 |
| 20141128_4d | n°1 | 311 ± 12 [318] | 30.4 | 284 ± 40 [361] | 25.0 | 251 | 1.60 | 30 14:56 | 0.14 | 36 ± 0 |
| | n°2 | 286 ± 28 [312] | 18.8 | 261 ± 41 [357] | 15.1 | 215 | 0.99 | 29 22:28 | 0.07 | 40 ± 4 |
| | n°5 | 222 ± 37 [264] | 20.9 | 234 ± 36 [356] | 20.7 | 606 | 1.99 | 30 07:45 | 0.67 | 58 ± 5 |
| | Outlet | 269 ± 61 [392] | 14.5 | 257 ± 54 [492] | 12.8 | 978 | 0.93 | - | - | 48 ± 10 |

Table 3: Main features of the selected flash flood events. Observed forcing PLU: network of 19 rain-gauges, observed forcing JP1: 1 km$^2$ quantitative precipitation estimates, Cumulated P (mm): mean ± standard deviation [max] accumulated precipitation on the catchment during the whole event, Max I (mm/h): mean of the maximal rainfall intensity over the catchment, $Q_p^o$ $(m^3/s)$: peak discharge for the event, $Q_p^o\big|_s$ $(m^3/s/km^2)$: ratio of the peak discharge for the event to the drainage area of the subcatchment, $T_p^o$ $(dd\ hh{:}mm)$: date of the peak discharge, $C_r$ (-): observed runoff coefficient, ratio of the amount of runoff through the outlet to the amount of rainfall on the catchment, $H_{ini}$ (%): mean ± standard deviation initial soil moisture according to SIM daily root-zone humidity output (Habets et al., 2008).

Figure 2 shows the spatial repartition of the cumulative rainfall for the three events for both forcings. The rain gauges data have been interpolated using the Thiessen polygon methods (Thiessen, 1911). Variability in rainfall clearly emerges especially between the eastern, western and mountainous part.

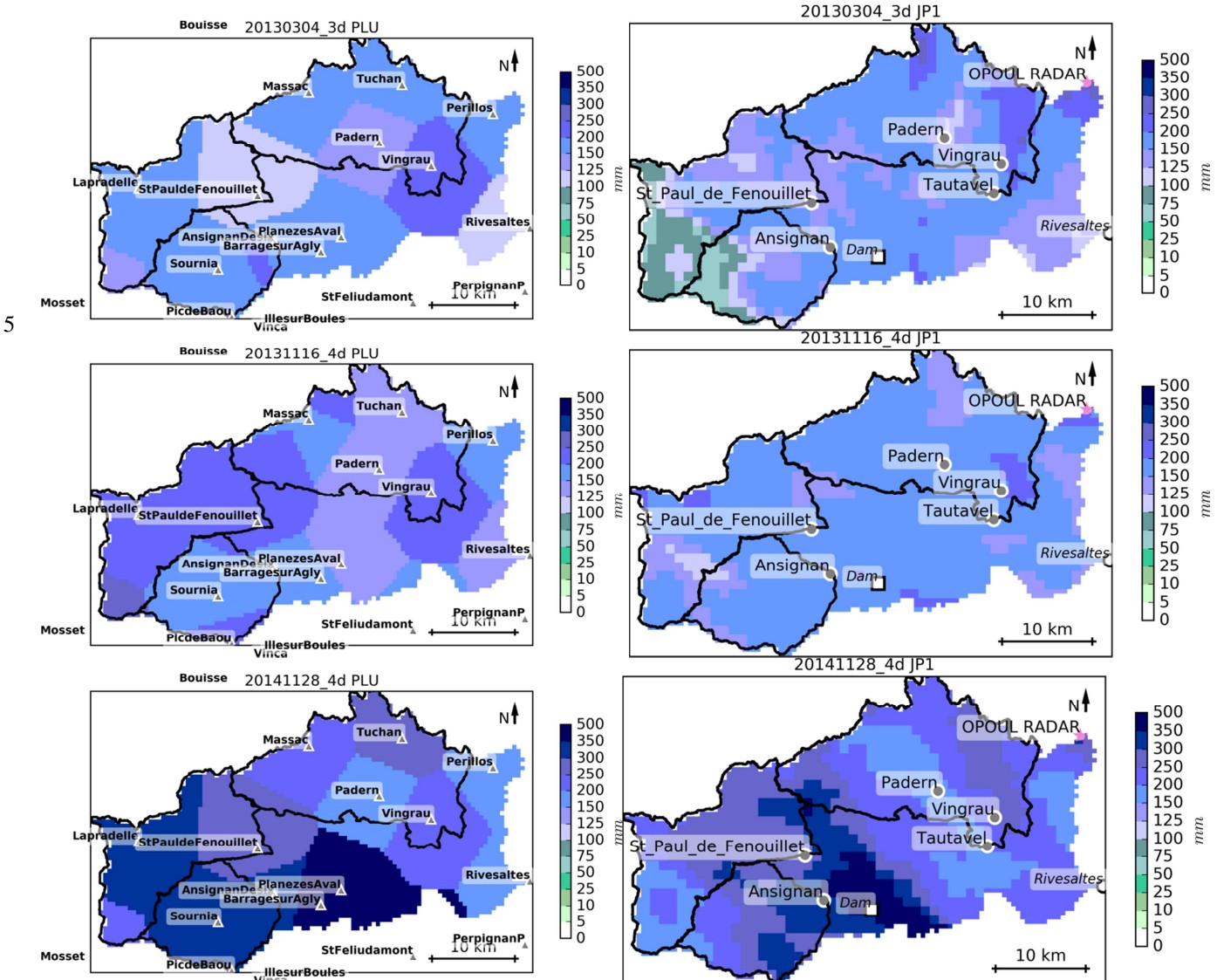

**Figure 2: Spatial variability of the cumulative rainfall for event 20130304_3d (top), 20131116_4d (middle) and 20141128_4d (bottom), according to the observations: PLU (left) the operational hourly rain-gauge network (from Hydroreel, Serveur de données hydrométriques en temps réel, Bassin Rhône-Méditerranée et Région Auvergne-Rhône-Alpes, https://www.rdbrmc.com/hydroreel2/listestation.php accessed on November 20,2019) and JP1 (right) 1 km² merging of radar data and rain-gauges measurements.**

## 3 Hydrological tool

### 3.1 Rainfall-runoff model

The MARINE model is a distributed mechanistic hydrological model specially developed for flash flood simulations. It models the main physical processes in flash flooding: infiltration, overland flow, lateral flows in soil and channel routing.
Conversely, it does not incorporate low-rate flow processes such as evapotranspiration or base flow.

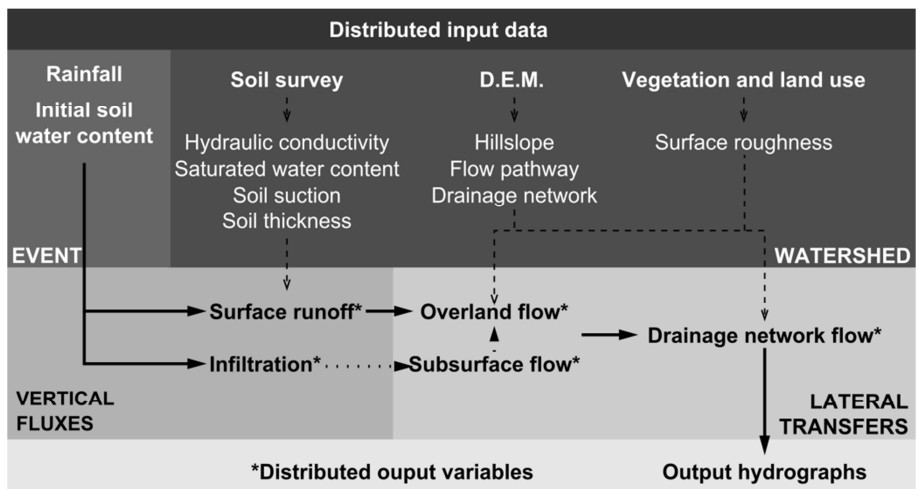

**Figure 3: Structure of the MARINE model.**

MARINE is structured into three main modules that are run for each catchment grid cell (Figure 3). The first module allows the separation of surface runoff and infiltration using the Green-Ampt model (Green and Ampt, 1911). The second module
represents subsurface downhill flow, based on the generalised Darcy law used in the TOPMODEL hydrological model (Beven and Kirby, 1979). Lastly, the third module represents overland and channel flows. Rainfall excess is transferred to the catchment outlet using the Saint-Venant equations simplified with kinematic wave assumptions (Fread, 1992). The model distinguishes grid cells with a drainage network –where channel flow is calculated on a triangular channel section (Maubourguet et al., 2007) – from grid cells on hillslopes, where overland flow is calculated for the entire surface area of the
cell. For more details about the MARINE model, the readers can refer to Roux et al. (2011), Garambois et al. (2015b) and Douinot et al. (2018).

The MARINE model works with distributed input data such as: (i) a digital elevation model (DEM) of the catchment to shape the flow pathway and distinguish hillslope cells from drainage network cells, according to a drained area threshold; (ii) soil survey data to initialize the hydraulic and storage properties of the soil, which are used as parameters in the infiltration
and lateral flow models; (iii) vegetation and land-use data to configure the surface roughness parameters used in the overland flow model. As the MARINE model is event-based, it must be initialized to take into account the previous moisture state of the catchment. This is done by using the spatial daily root-zone saturation state, i.e. the ratio of the soil water content to the soil storage capacity at a spatial resolution of 8×8 km, output from Météo-France's SIM operational chain (Habets et al.,

2008). The initial soil water content for MARINE is therefore directly obtained by multiplying the saturation state by the soil storage capacity of each cell.

## 3.2    Calibration/Validation on the Agly catchment

MARINE requires parameter calibration so as to accurately reproduce hydrological behaviours. Based on previous sensitivity analyses by Garambois et al. (2013), five parameters are calibrated: soil depth $C_Z$, the transmissivity used in lateral subsurface flow modelling $C_T$, hydraulic conductivity at saturation $C_K$, and friction coefficients for low and high-water channels, $n_L$ and $n_H$, respectively. $C_T$, $C_K$ and $C_Z$ are the multiplier coefficients for spatialised, saturated hydraulic conductivities and soil depths. Note that $n_L$ and $n_H$ are kept invariant throughout the drainage network. The spatial resolution of the MARINE model on all the Agly subcatchments is of $500\ m$. The calibration of the Agly catchment at the Saint-Paul-de-Fenouillet station (n°2, Table 1 and Figure 1) were performed by Garambois et al. (2015a) according to their proposed methodology. The events used for this calibration are older than those considered in the present study (20020411, 20031204, 20040221, 20051115, 20101010, 20110315, see Garambois et al., 2015a). The cost function $L_{NP}$ is designed to evaluate the performance of the model (Roux et al., 2011; Garambois et al., 2015a):

$$L_{NP} = \frac{1}{3}L_N + \frac{1}{3}\left(1 - \frac{|Q_p^s - Q_p^o|}{Q_p^o}\right) + \frac{1}{3}\left(1 - \frac{|T_p^s - T_p^o|}{T_c}\right), \tag{1}$$

where $Q_p^s$ and $Q_p^o$ are respectively the simulated and observed peak runoff, $T_p^s$ and $T_p^o$ are the simulated and observed time to peak, and $T_c$ is the time of concentration of the catchment. $L_N$ denotes the efficiency coefficient (Nash and Sutcliffe, 1970):

$$L_N = 1 - \frac{\sum_{i=1}^{n}(Q_i^s - Q_i^o)^2}{\sum_{i=1}^{n}(Q_i^o - \overline{Q^o})^2}, \tag{2}$$

where $n$ is the number of observation data, and $Q^s$ and $Q^o$ are the simulated and the observed runoff. The estimated times of concentration of each subcatchment are given in Table 1, using Bransby Williams formula (Pilgrim and Cordery, 1992):

$$T_c = 14.6 L A^{-0.1} S^{-0.2}, \tag{3}$$

where $T_c\ [min]$ is the time of concentration, $L\ [km]$ is the total length of channel, $A\ [km^2]$ is the drainage basin area, $S\ [m/m]$ is the average slope. Here, the formula for time of concentration is only used to normalize the peak time delay in the third term of equation 1 with a characteristic time of the catchment, so the most important point is to always use the same procedure to make this term dimensionless. Note that the range of values for both $L_{NP}$ and $L_N$ spans from $-\infty$ to 1, one being the perfect score.

Table 4 lists the $L_N$ and $L_{NP}$ efficiencies for the validation cases: the 3 studied events with different forcings and 2 older flash flood events with available data, only used for the validation process of the hydrological model, but not further studied. Table 4 and Figure 4 show that:

- Only one event (20130304_3d with PLU forcing) is well simulated at the 5 gauging stations,

- Only one event (20130304_3d with both PLU and JP1 forcings) is well simulated at mountainous station n°1,

- All the other events are correctly simulated only for a part of the catchment: either the eastern part near the Mediterranean Sea (stations n°3, n°4 and n°5), or the south-west mountainous part (station n°1), or the north-west continental part (station n°2). This result doesn't seem to be directly linked with the rain-gauged distribution because first of all, the rain-gauge network is quite dense in this catchment and rather well distributed: with 19 rain-gauges for an area of around 1000 $km^2$, the rain-gauges density is about 1 for 50 $km^2$ whereas the rain-gauge density for the full network over mainland France is of 1 for 120 $km^2$ (Mounier et al., 2012). In addition, it's not always for the same part of the catchment that the model has the best performance: it depends on the event. Therefore, the same distribution of rain-gauges sometimes leads to a correct simulation in term of $L_{NP}$ cost function (Eq. 1) for a given even, while leads to an unsatisfactory simulation for another event.

| Event forcing | n°1 | n°2 | n°3 | n°4 | n°5 |
|---|---|---|---|---|---|
| 19920926_PLU | - | **0.92(0.93)** | - | - | |
| 20090411_PLU | <0(<0) | **0.50**(0.12) | <0(<0) | - | <0(<0) |
| 20130304_3d_PLU | **0.78(0.80)** | **0.61**(0.72) | **0.61**(0.43) | **0.67**(0.60) | **0.70**(0.61) |
| 20130304_3d _JP1 | **0.74(0.73)** | <0(0.34) | **0.67**(0.52) | **0.77**(0.66) | **0.78**(0.69) |
| 20131116_4d _PLU | <0(<0) | **0.64**(0.41) | 0.06(<0) | <0(<0) | 0.38(<0) |
| 20131116_4d _JP1 | <0(<0) | <0(0.36) | <0(<0) | <0(<0) | 0.24(<0) |
| 20141128_4d _PLU | <0(<0) | 0.11(<0) | **0.65**(0.16) | **0.67**(0.47) | **0.79**(0.61) |
| 20141128_4d _JP1 | <0(<0) | **0.68(0.64)** | **0.78(0.73)** | **0.81(0.74)** | **0.89(0.81)** |

**Table 4: $L_{NP}(L_N)$ efficiencies for each station (see numbering Table 1) and for each validation events, PLU: forcing with the network of 19 raingages, JP1: forcing with 1 $km^2$ quantitative precipitation estimates. Bold values indicate efficiencies above 0.5.**

As expected, the different parts of the catchment exhibit various behaviours which are difficult to correctly simulate with a single calibration by just using observations at the station n°2. On one hand, events with relatively moderate peak discharge are usually not correctly simulated by MARINE whatever the observed forcing, as is the case of the 20090411_PLU and 20131116_4d events. Indeed, several authors have pointed out that specific peak discharges larger than 0.5 $m^3 s^{-1} km^{-2}$ are one of the relevant criteria to define a flash flood (Braud et al., 2014; Gaume et al. 2009). The 20090411_PLU and 20131116_4d events exhibit smaller peak discharges (Table 3), except for the 20131116_4d episode at station n°2, where the results are correct for the PLU forcing (Figure 4). When the simulated hydrographs are suitable for the eastern Agly, the discharge is overestimated over the western part (e.g. 20141128_4d; Figure 4). Conversely, when the simulated hydrographs are correct over the western Agly, the peak discharges are underestimated in the eastern part as in the 20130304_3d episode. Difficulties in correctly simulating the hydrological responses over all the subcatchments arise due to

the spatial variability of hydrological behaviour across the Agly catchment, leading to a myriad of runoff responses that are difficult to encompass with single parameterizations of the infiltration process in hydrological models (Amengual et al., 2017).

With respect to the two major 20130304_3d and 20141128_4d events, both simulated with the two observed forcing, simulations are more satisfactory with the 1 km² quantitative precipitation estimates ANTILOPE J+1 for the eastern than for the western part. This may be due to the fact that the radar is located close to the sea, being the beams orographically sheltered over the western Agly (Figure 1). Several other calibration tests could have been carried out so as to improve the results of the hydrological model such as one calibration for each sub-catchment. However, the main purpose of this study focuses on the potential of ensemble strategies to improve flash flood forecasting. Furthermore, NWP model driven runoff simulations have been compared both against the observed discharges and against the observed rain-gauge and radar precipitation driven runoff runs. Hence, the impact of the external-scale uncertainties on the quality of the distinct HEPS can be emphasized.

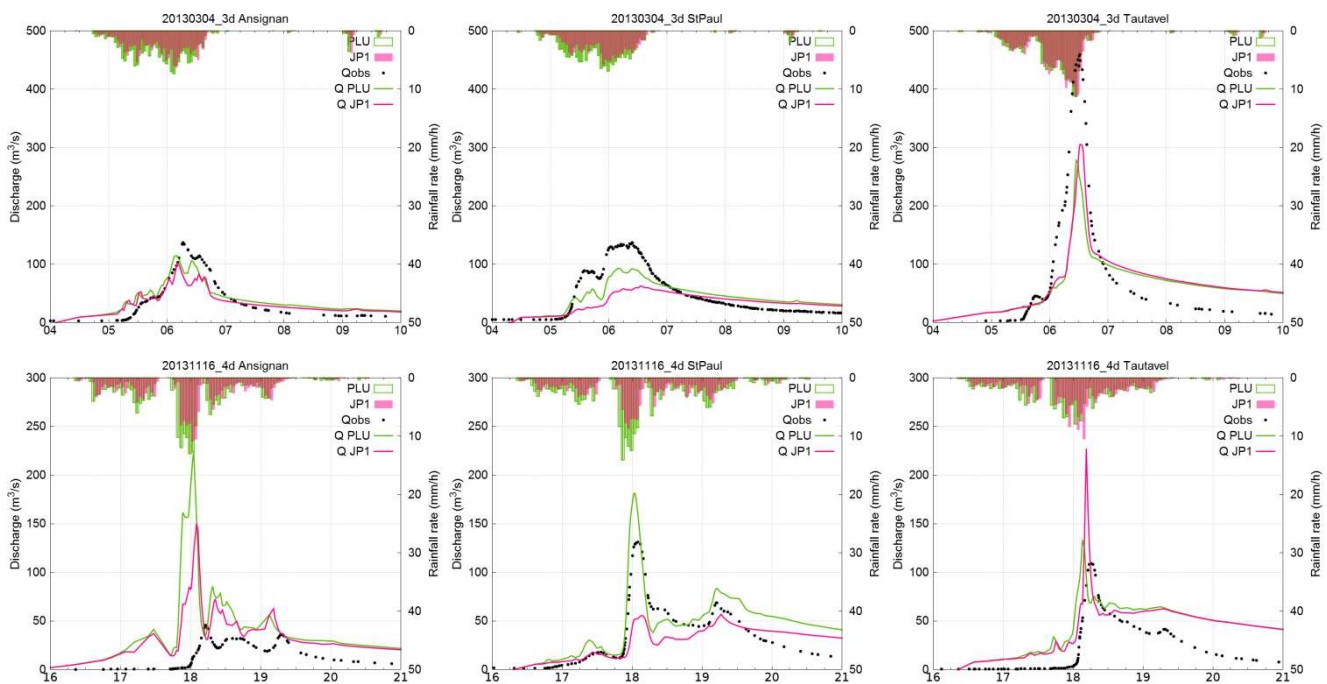

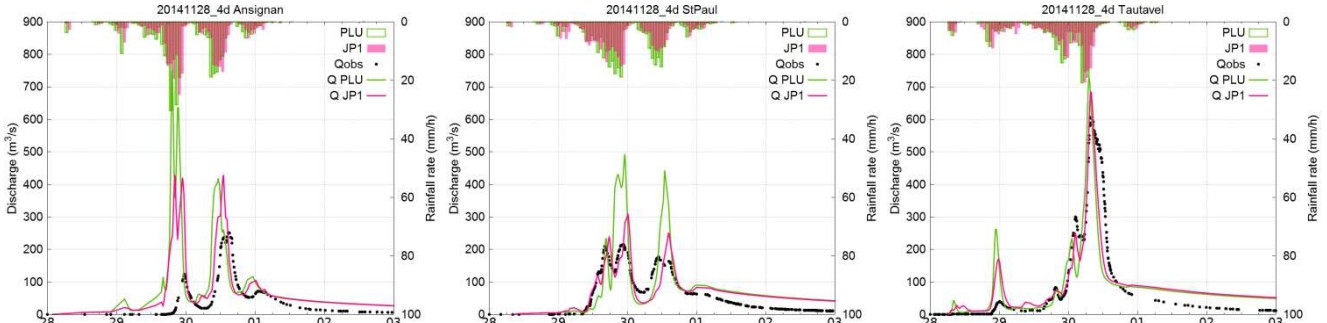

**Figure 4 : Hyetogram and hydrogram at station n°1 (left), n°2 (center) and n°5 (right) for three events, PLU: forcing with the network of 19 raingages, JP1: forcing with 1 km² quantitative precipitation estimates, Qobs: observed discharge at the station, Q PLU: simulated discharge with PLU forcing, Q JP1: simulated discharge with JP1 forcing.**

## 4   Meteorological tools

The fully compressible and non-hydrostatic WRF model has been employed to generate the ensemble members. The WRF set-up consists of a single computational domain completely spanning the Western Mediterranean region at 2.5 km spatial horizontal resolution (i.e. 767 x 575 grid-points) and 50 vertical levels (Figure 5). Deep moist convection is explicitly solved due to the high-spatial resolution. All the ensemble experiments have a temporal forecasting horizon of 48-h, starting at 00 UTC on the day before of the main observed peak floods. Starting on this day warranties a suitable lead-time to issue warnings to local water management services. For these hydro-meteorological episodes lasting more than 2 days, successive consecutive 48-h simulations have been performed, starting on the next days at 00 UTC. Hence, the initiation and subsequent evolution of the most active precipitation systems and the overall rainfall episodes are completely encompassed.

WRF simulations have been forced by using the global Ensemble Prediction System of the European Centre for Medium Range Weather Forecasts (ECMWF-EPS). The MPS ensemble has been built by using the reference (i.e. unperturbed) run, while the PILB approach has considered a selected set of the overall ECMWF-EPS population. Finally, the hourly QPFs are used to force one-way the MARINE model so as to build the HEPSs. In addition, the deterministic ECMWF forecasts have been also dynamically downscaled so as to have a control baseline for comparative purposes against the ensemble strategies.

Deterministic simulations have used the following physical parameterizations: the WRF single moment 6-class microphysics scheme, including graupel (WSM6; Hong and Lim 2006); the 1.5-order Mellor–Yamada–Janjić boundary layer scheme (MYJ; Janjić, 1994); the Dudhia short-wave scheme (Dudhia, 1989); the RRTM longwave scheme (Mlawer et al., 1997); the unified Noah land surface model (Tewari et al. 2004); and the Eta similarity surface-layer model (Janjić, 1994). Note that the WRF configuration for the control simulations is the same as the daily operational set-up run by the research Meteorology Group at the University of the Balearic Islands (http://meteo.uib.es/wrf).

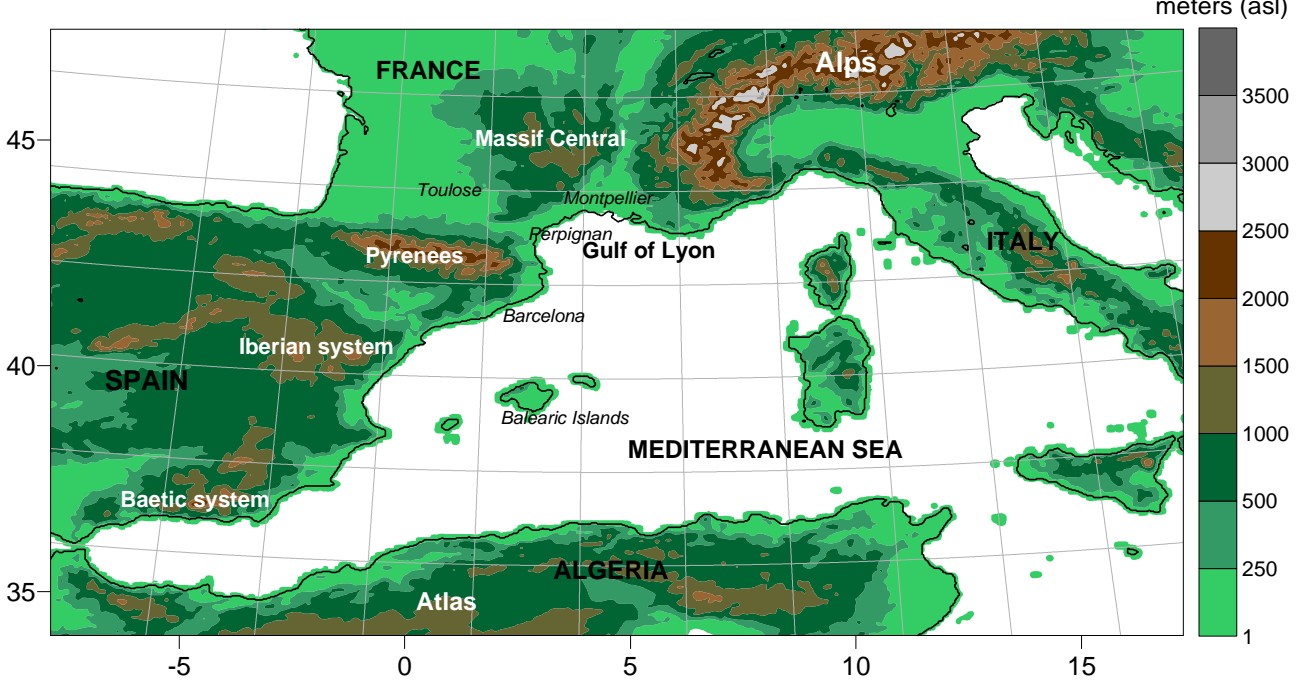

**Figure 5 : Configuration of the computational domain used for the WRF numerical simulations.**

### 4.1    PILB ensemble

The operational ECMWF-EPS is formed by 51 members –the reference and 50 perturbed forecasts– at T639 spectral resolution (20 km) and aims to cope with uncertainties related to the actual state of the atmosphere. The daily synoptic-scale uncertainties are encompassed by perturbing an initial analysis through the flow-dependent singular vectors technique (Buizza and Palmer, 1995; Molteni et al., 1996). However, perturbed IC/LBCs can produce inadequate spread in the short range, before error growth on the synoptic scale becomes non-linear (Gilmour et al., 2001). Therefore, the implemented

PILB ensemble is based on dynamically downscaling these 20 ECMWF-EPS members exhibiting maximum perturbations in the initial and lateral boundaries conditions over the WRF domain. This strategy seeks to ameliorate the aforementioned mismatch between the synoptic-scale error growth optimization time for the singular vectors and the sub-synoptic error growth, more relevant for short-range forecasts at small- and medium-sized basins (Ravazzani et al., 2016; Amengual et al. 2017).

At this aim, a k-means clustering algorithm using the Principal Components of the 500 hPa geopotential and 850 hPa temperature fields is applied to the entire ECMWF-EPS over the WRF numerical domain. Then, the 50 ensemble members are categorized in 20 clusters and the 20 closest members to the centroids are used as initial and boundary fields for the PILB

ensemble. Boundary fields are updated every 3 h and physical schemes remain invariant for all the ensemble members and are the same that these used to run the deterministic WRF simulations.

## 4.2    Mixed-physics (MPS) ensemble

There is not an optimum set of physical numerical parameterizations when simulating severe weather and intense precipitation events. Several studies have shown that different combinations of physical parameterizations render similar performances (Jankov et al., 2005; Evans et al., 2012). That is, the meteorological variables are sensitive to a myriad of processes which are differently parameterized by capable numerical schemes. When simulating flash flooding driven by convective-type precipitation, cumulus parameterizations are the main candidates for direct uncertainty sampling. However, as convection is explicitly resolved, uncertainties arising from the microphysical sub-grid processes and planetary boundary layer (PBL) schemes have been encompassed. The former regulates the distinct forms of rainfall, the latter accounts for the turbulent vertical fluxes of heat, momentum and moisture within the PBL and throughout the atmosphere. Both physical mechanisms are also dominant when controlling deep moist convection. The MPS ensemble has been generated using all possible pairs (cloud microphysics-boundary layer) between the following schemes, summing up to 20 members:

-   Microphysical schemes: (i) WRF single-moment 6-class (WSM6; Hong and Lim, 2006); (ii) Goddard (Tao et al., 1989); (iii) New Thompson (Thompson et al., 2008); and (iv, v) National Severe Storm Laboratory (NSSL) two-moment (Mansell et al., 2010) with two Cloud Condensation Nuclei (CCN) prediction values of $0.5 \cdot 10^9$ and $1.0 \cdot 10^9$ $cm^{-3}$.

-   PBL schemes: (i) Yonsei University (YSU; Hong et al., 2006); (ii) Mellor-Yamada-Janjic (MYJ; Janjic, 1994); (iii) Mellor–Yamada–Nakanishi–Niino level 2.5 (MYNN; Nakanishi and Niino, 2006)), and (iv) Total Energy–Mass Flux (TEMF; Angevine et al., 2010).

On one hand, all microphysics schemes involve the simulation of explicitly resolved liquid water, cloud and precipitation, and include mixed-phase transformations (i.e. the interaction of ice and liquid water). However, each microphysical parameterization treats differently the interaction among five or six moisture species (i.e. water vapour, cloud water, rain, cloud ice, snow and graupel); the physical processes of rain production, fall and evaporation; the cloud water accretion and auto-conversion; condensation; and saturation adjustment and ice sedimentation. The Western Mediterranean is affected by air masses of distinct signature (i.e. Saharan, Atlantic, purely Mediterranean or continental central European), featuring a high variability of aerosol concentration that influence the moist physical mechanisms. The inclusion of two different CCN concentrations copes with uncertainties in the aerosol characteristics. On the other hand, the choice of different PBL schemes can be crucial when correctly simulating the onset of mesoscale severe weather phenomena. PBL modulates the temperature and moisture profiles in the lower troposphere and the effects of turbulence in the daytime convective conditions (Hu et al. 2010; Coniglio et al. 2013). Finally, it is worth noting that the initial and lateral boundary

conditions are kept invariant through all the MPS ensemble members. IC/LBC come from the ECMWF-EPS reference forecast for each individual case study and lateral boundary conditions are updated every 3 h.

## 5    Results and discussion

### 5.1    Verification of the SREPS

The quantitative comparison of the spatial 48-h accumulated precipitations for the PILB and MPS experiments against the radar estimates provides a quality outlook of the ensemble performance for the selected episodes over the study region. Figure 6, Figure 7 and Figure 8 indicate realistic spatial distributions for all the study cases: high rainfall accumulations in the upper tail distributions of both ensemble strategies are a good indication of the potential for heavy rainfall. The regional roughed topography (i.e., the pre-Pyrenees, Pyrenees and the Massif Central) is determinant to place and focus the probabilistic quantitative precipitation forecasts. Both approaches could succeed for issuing warning alerts before flash flood scenarios in the region. However, SREPS reliability must be previously checked at basin scales. Flash flood forecasting over a single medium-sized catchment is a challenging issue as many small-scale atmospheric factors concur in determining the location of deep convection and intense precipitation. A crucial feature in determining correctly the location of the rainfall amounts is to accurately simulate the south to north easterly low-level moisture maritime flows impinging over the mountainous slopes of the Agly basin.

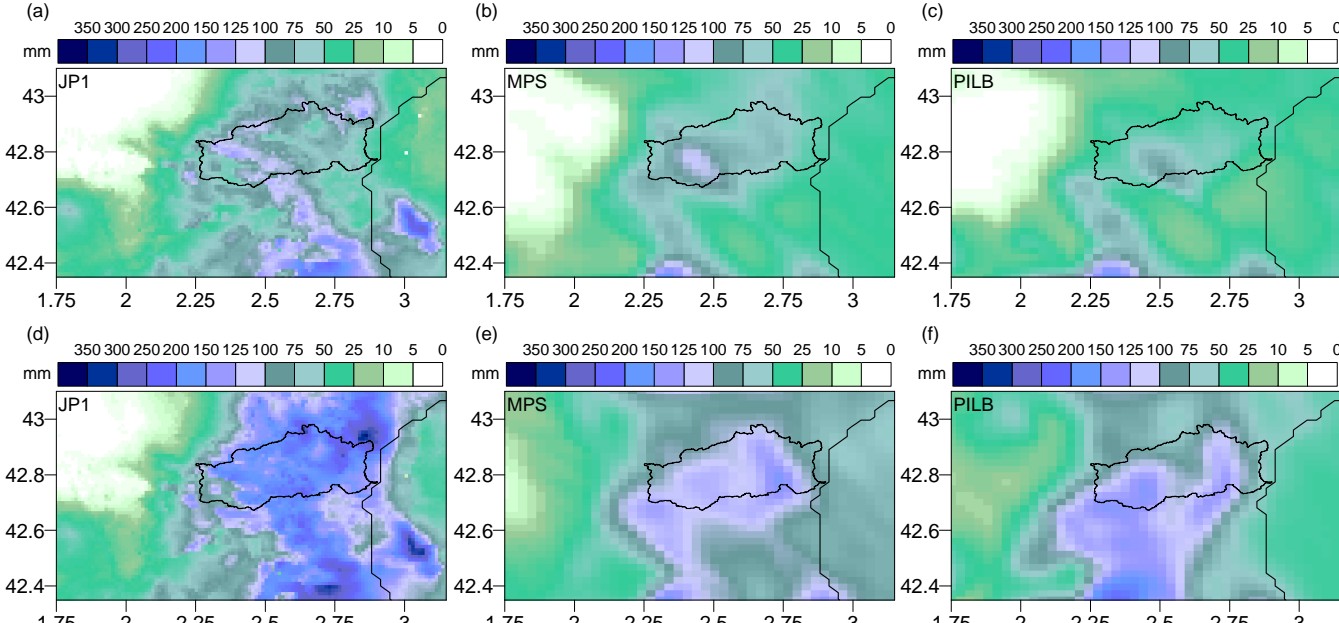

**Figure 6 : Spatial distributions of the 48-h rainfall amounts for the March 2013 episode according to: (a) radar JP1, (b) MPS percentile 90 and (c) PILB percentile 90, starting on 4th 00 UTC, and; (d) radar JP1, (e) MPS percentile 90 and (f) PILB percentile 90, starting on 5th 00 UTC. The Agly basin is highlighted.**

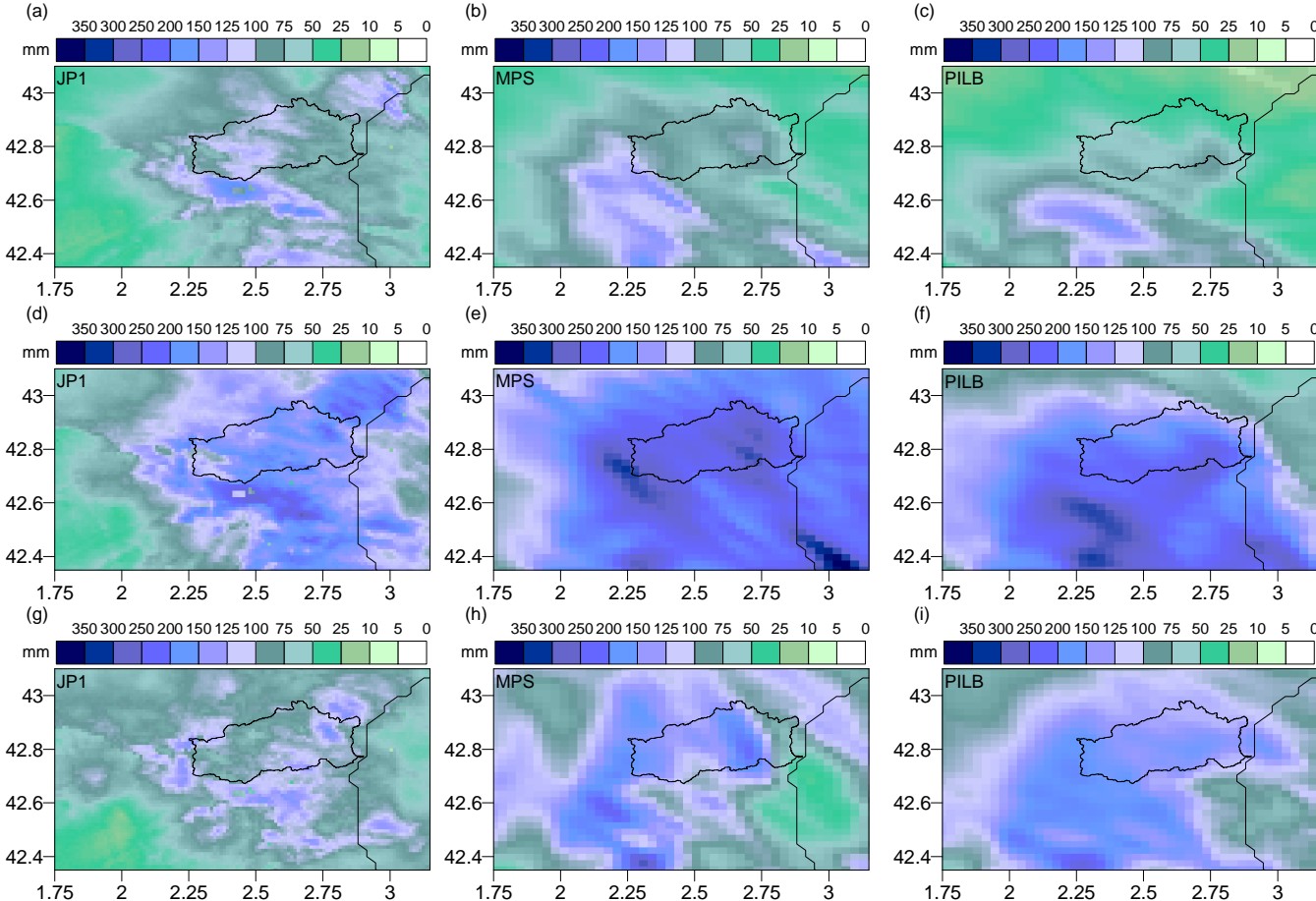

**Figure 7 : Spatial distributions of the 48-h rainfall amounts for the November 2013 episode according to: (a) radar JP1, (b) MPS percentile 90 and (c) PILB percentile 90, starting on 16th 00 UTC; (d) radar JP1, (e) MPS percentile 90 and (f) PILB percentile 90, starting on 17th 00 UTC; and (g) radar JP1, (h) MPS and (i) PILB, starting on 18th 00 UTC. The Agly basin is highlighted.**

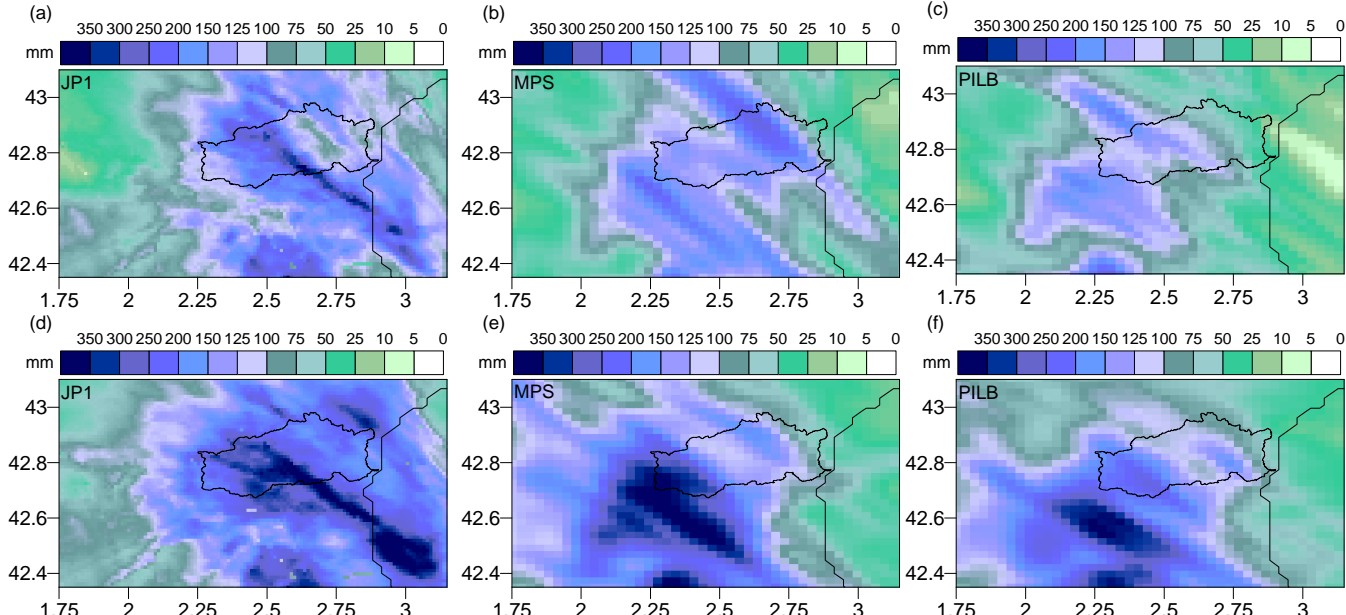

**Figure 8 : Spatial distributions of the 48-h rainfall amounts for the November 2014 episode according to: (a) radar JP1, (b) MPS percentile 90 and (c) PILB percentile 90, starting on 28th 00 UTC; and (d) radar JP1, (e) MPS percentile 90 and (f) PILB percentile 90, starting on 29th 00 UTC. The Agly basin is highlighted.**

48-h rain-gauge (PLU) and radar-derived (JP1) rainfall amounts have been used to evaluate the forecasting ensemble skill at the relevant hydrological scales. To this end, the cumulative ensemble QPFs have been interpolated to all the available rain-gauges and to the pixels of the radar domain shown in Figures 6 to 8 for each study case (Akima, 1978 and 1996; Figure 9). Most members of the PILB and MPS ensembles exhibit underestimations for the 04-05/03/2013 and 28-29/11/2014 experiments, while overestimations for the 16-18/11/2013 simulations. Both strategies do not present remarkable differences in ensemble skill and spread when forecasting the total rainfall amounts (Figure 10). Root mean squared errors (RMSE) and correlations (r) are quite similar, indicating a slightly more accurate performance of the MPS or PILB ensemble strategy depending on the case study and the starting day of the experiment.

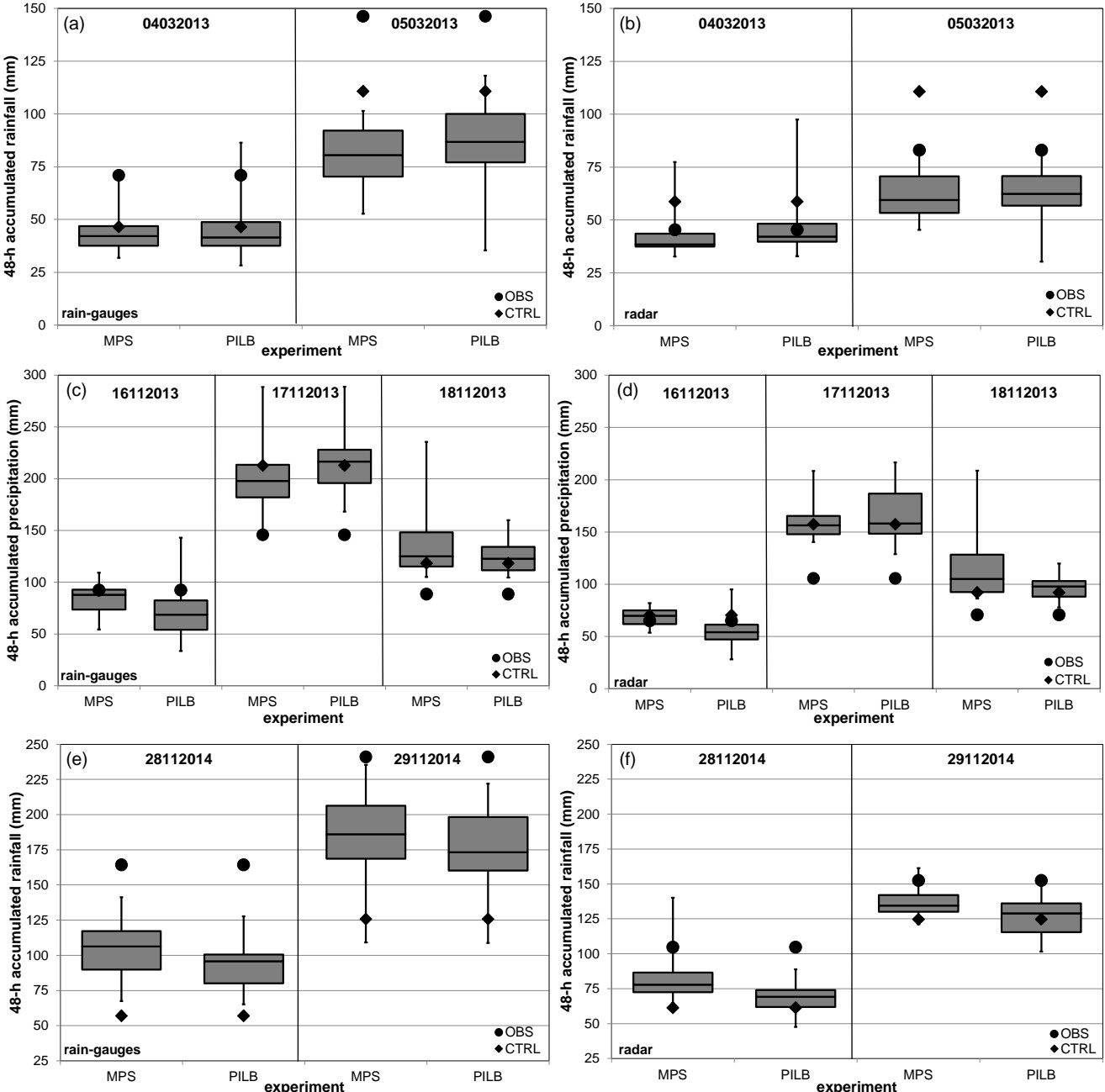

**Figure 9 : 48-h rainfall amounts according to the rain-gauge (PLU, left) and radar-derived (JP1, right) observations and the PILB and MPS experiments. Boxes denote the p25 and p75 interquartile ranges, middle horizontal lines show the ensemble median and whiskers display the tails of the ensemble. Note that the PILB and MPS ensemble experiments start on the day indicated in the upper part of each subpanel. CTRL stands for the control or deterministic simulation.**

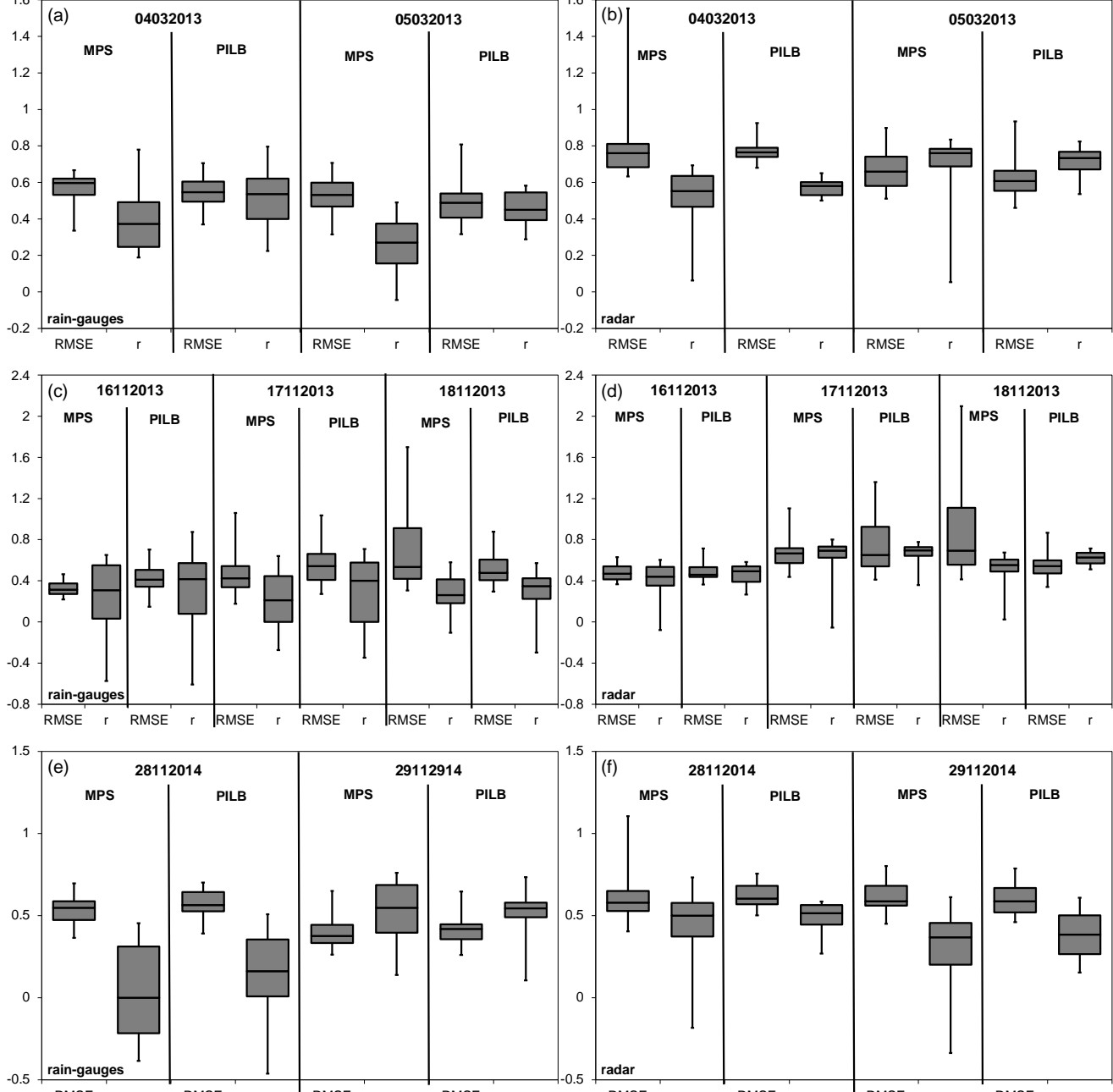

**Figure 10 : Statistical scores of the 48-h rainfall amounts for the PILB and MPS ensemble members when compared against the rain-gauge (PLU, left) and the radar-driven (JP1, right) observations. Boxes denote the p25 and p75 interquartile ranges, middle horizontal lines show the ensemble median and whiskers display the best and the worst ensemble members. Note that the PILB and MPS ensembles start on the day indicated in the upper part of each subpanel.**

In addition, the skill of each ensemble strategy in predicting the probability for different accumulations –ranging from light to torrential rainfalls– has been assessed by means of the ROC curves. The ROC curve expresses the true hit rate of a probabilistic forecast at different false alarm rates, while the area under the ROC curve (AUC) quantifies the ability of the ensemble to discriminate between the occurrence or non-occurrence of an event (Schwartz et al., 2010). ROC curves have been computed by using all the study cases and the radar-derived (JP1) rainfall accumulations have been employed as the observed baseline. The following 48-h accumulated precipitation thresholds have been considered: 5, 10, 15, 25, 50, 75, 100, 125, 150 and 200 mm. As the forecast probabilities are computed and verified against each pixel within the radar domain shown in Figures 6 to 8, the statistical sample sums up to 54145 members (7735 radar grid-points times 7 ensemble experiments).

Probabilistic QPFs from the PILB approach shows slightly higher forecasting skills than MPS for small rainfall accumulations (i.e., $\leq 15$ mm; Table 5 and Figure 11). Even so, the AUCs are above 0.85 for both ensemble strategies. For moderate to high rainfall thresholds (25-75 mm), PILB and MPS are almost statistically indistinguishable, with AUCs well above 0.7. Depending on the precipitation limit, MPS or PILB features a slightly higher probabilistic forecasting skill. At greater thresholds ($\geq 100$ mm), PILB shows a larger discrimination ability, with areas slighter higher than 0.7 for all the cases, except the most extreme precipitation accumulation. On the other hand, MPS renders values close to but below 0.7. In general, both strategies exhibit an elevate quality of the probabilistic forecasts for low to moderate rainfall accumulations. Remarkably, the discrimination ability of the PILB strategy is maintained up to 150 mm. This result points out to a more effective encompassing of uncertainties emerging from the IC/LBCs than from the microphysical and PBL physical inaccuracies likely due to the dominant role of the regional complex orography when controlling rainfall location. However, the high AUCs rendered by both ensemble strategies suggest to account for both sources of uncertainty so as to obtain high-quality probabilistic quantitative precipitation forecasts.

| Precipitation threshold (mm) | ROC areas | |
|---|---|---|
| | MPS | PILB |
| 5 | 0.855 (0.846–0.864) | 0.917 (0.911–0.922) |
| 10 | 0.888 (0.881–0.894) | 0.913 (0.909–0.917) |
| 15 | 0.852 (0.846–0.859) | 0.877 (0.872–0.881) |
| 25 | 0.833 (0.828–0.839) | 0.842 (0.837–0.847) |
| 50 | 0.785 (0.780–0.790) | 0.771 (0.766–0.776) |
| 75 | 0.741 (0.735–0.746) | 0.741 (0.736–0.747) |
| 100 | 0.699 (0.694–0.705) | 0.721 (0.715–0.726) |
| 125 | 0.690 (0.684–0.695) | 0.717 (0.711–0.722) |
| 150 | 0.691 (0.685–0.697) | 0.716 (0.710–0.721) |
| 200 | 0.638 (0.630–0.647) | 0.689 (0.682–0.696) |

**Table 5 : Areas under the ROC curves for the MPS and PILB ensemble strategies. Associated uncertainty to each score (between brackets) is expressed as the 95% percentile confidence intervals, calculated by using a 10000-sample bootstrap.**

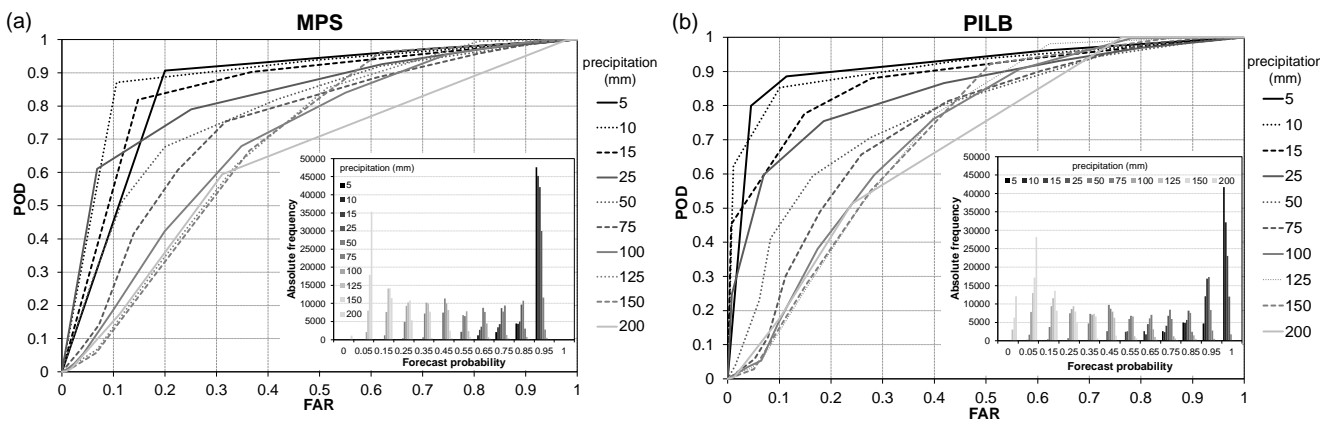

**Figure 11 : ROC curves of the MPS and PILB ensemble strategies. The embedded figures display the sharpness diagrams containing the number of forecasts used in each probability bin and the total number of observations considered.**

### 5.2    Verification of stream flow forecasts

As mentioned by Bellier et al. (2017), the visual inspection of individual hydrographs is useful for a better understanding of how forecasts behave. The hydrological simulations have been forced by the 48-h meteorological simulations, resulting in 7

hydro-meteorological simulations each lasting 2 days, starting respectively on the 4[th] and 5[th] of March 2013 (20130304_2d and 20130305_2d), 16[th], 17[th] and 18[th] of November 2013 (20131116_2d, 2013117_2d and 20131118_2d), 28[th] and 29[th] of November 2014 (20141128_2d and 20141129_2d) at 00 UTC. Figure 12 shows the hydrographs at three stations (n°1, n°2 and n°5) of the 20130305_2d, 20131117_2d and 20141129_2d experiments and for the all 48-h performed simulations with: observed forcing (PLU and JP1), deterministic (WRF) and ensemble forecast MPS. Results are very similar for PILB-HEPS.

The median and the 10[th] and 90[th] quantiles of each ensemble strategy, as well as the first level alert from the flood warning center in France (SCHAPI), are also shown as references. In general, the WRF deterministic driven hydrological forecasts often miss the peak times for all the hydrometric stations (Figure 12). The HEPS improves this feature, even if biases in the EPS still remain as are propagated down to the hydrological model. That is, the MPS-HEPS and PILB-HEPS exhibit slight underestimations (overestimations) for the 20130305_2d and 20141129_2d (20131117_2d) simulations. The observed peak

time is included in the boxplots (minimum and maximum of all of the data) of the ensemble strategies for the 5 stations, whereas it is not included in the boxplot for the deterministic simulations at stations n°1, 2 and 3 as it can be seen in Figure 13 for stations n°1 and 5. It can also be appreciated that the peak timing delay is usually negative, independently of the experimental set-up. Almost all the hydro-meteorological simulations result in earlier peak timings than observed.

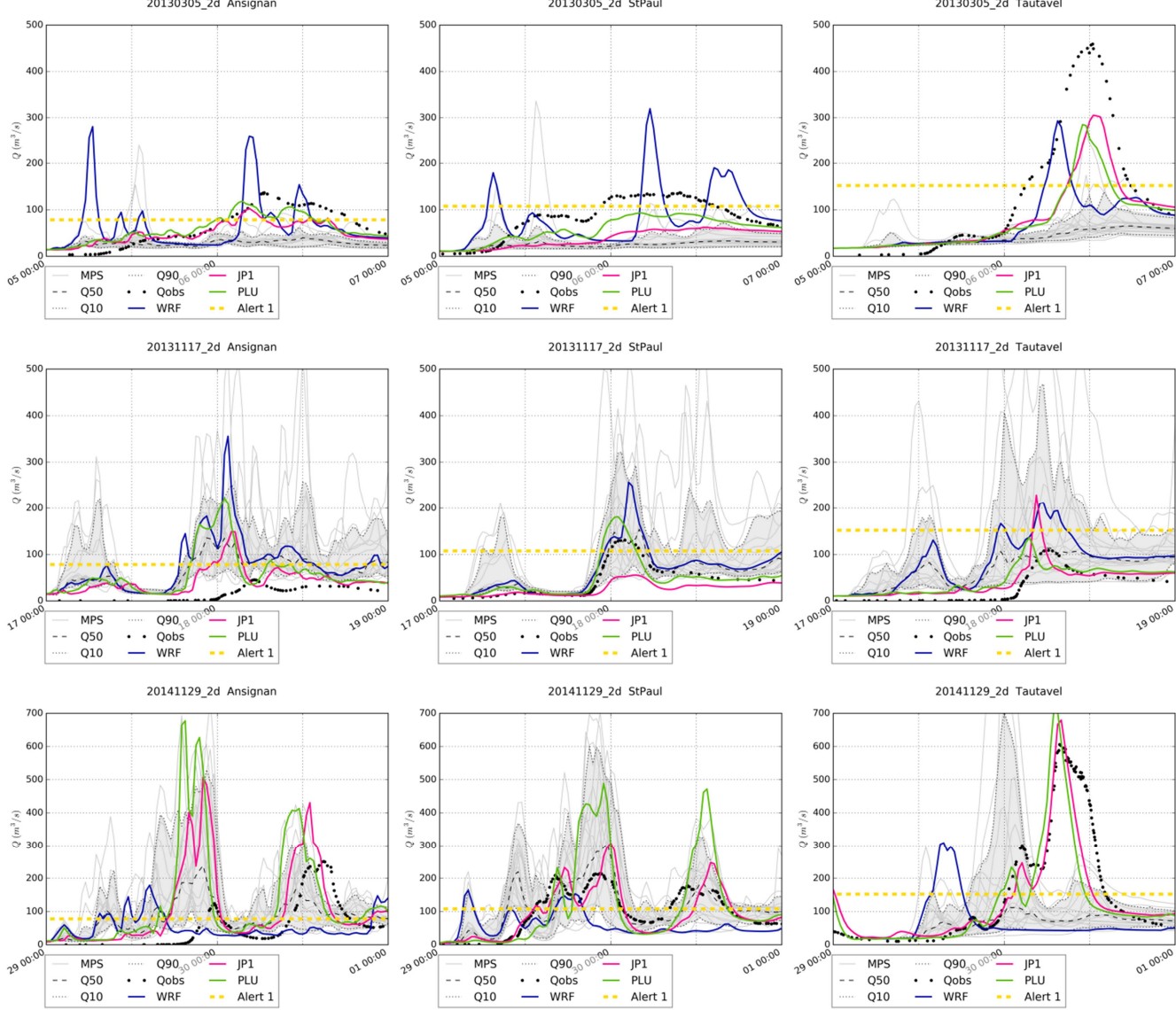

**Figure 12: MPS-HEPS hydrograms at station n°1 (left), n°2 (center) and n°5 (right) for the 20130305_2d simulation (top), 20131117_2d simulation (middle), 20141129_2d simulation (bottom). Note that Q50 is the ensemble median, Q10 denotes the 10th ensemble quantile, Q90 labels the 90th ensemble quantile, Qobs is the observed discharge, WRF is the WRF deterministic driven discharge experiment, PLU is the PLU driven runoff simulation, and JP1 denotes the JP1 driven discharge simulation. Alert 1 corresponds to the first alert level.**

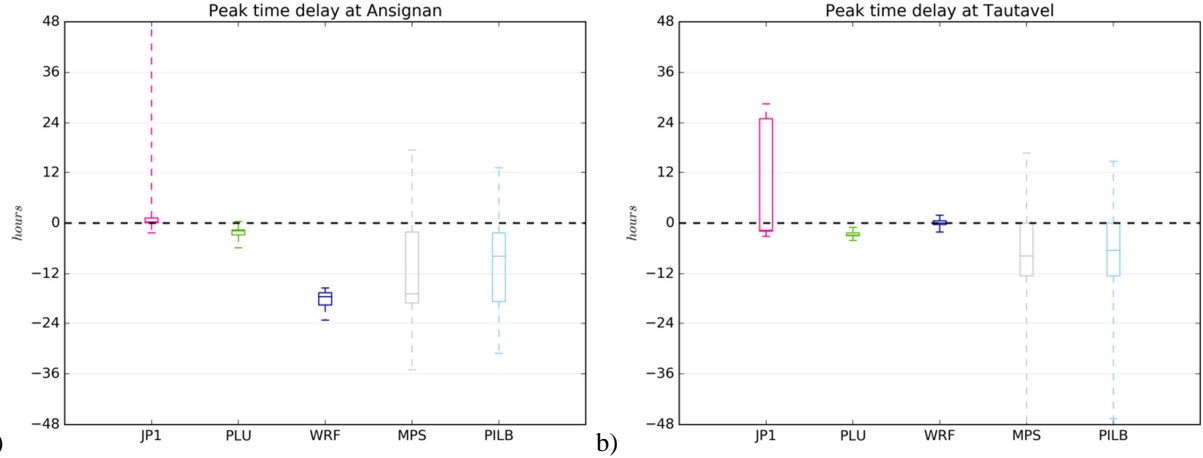

**Figure 13: Delay of simulated peak time for the 7 simulations at stations n°1 a) and n°5 b) for simulations with JP1 forcing, PLU forcing, WRF deterministic forcing and ensemble strategies forcings (MPS and PILB). The boxplot presents five sample statistics: the minimum, the lower quartile, the median, the upper quartile and the maximum.**

The peak plot approach has been adopted to better appreciate the value of the ensemble strategies: all the ensemble members are joined in a single plot by calculating the deviation from the observed peak discharge and timing (Zappa et al., 2013; Ravazzani et al., 2016). Figure 14, Figure 15 and Figure 16 summarize the simulations carried out for stations n°2 and n°5 and for simulations 20130305_2d, 20131117_2d and 20141129_2d. Results exhibit a high inter-event variability as it might be expected given their different characteristics. Regarding the MPS-HEPS experiments, the observed peak lies in the range

of variation of the ensemble for the 20130305_2d run at hydrometric stations n°1 and n°2 (Figure 14). This fact can be ascribed to the large spread found in the driven peak discharges: deviations from the observation range from approximately $-110$ to $+200\ m^3s^{-1}$, while timing delays fluctuate from $-26$ to $+15\ h$ for station n°2. Indeed, the 80% confidence interval of the MPS-HEPS simulations never encompasses the observed discharge for this event. The same remarks also apply for the 20141129_2d case at stations n°3, 4 and 5 (Figure 16) and 20131117_2d at station n°3. The 80% confidence

interval of the MPS-HEPS simulations encompasses the observed discharge only for the 20131117_2d simulation at stations n°2, 4 and 5 (Figure 15) and for the 20141128_2d at station n°2.

The observed peak also lies in the range of variation of the PILB-HEPS ensemble strategy for the 20131117_2d run at stations n°2, 3, 4 and 5 (Figure 15), and for the 20141129_2d simulation at the five gauge-stations (Figure 16). Concerning both episodes at the gauge-station n°2, PILB-HEPS spread is larger than MPS-HEPS in terms of the observed peak discharge

although smaller for the observed peak time. That is, from $-17$ to $+22\ h$ for the MPS-HEPS and from $-3\ h$ to $+18\ h$ for the PILB-HEPS for 20131117_2d and from $-12$ to $+25\ h$ for the MPS-HEPS and from $-12\ h$ to $+8\ h$ for the PILB-HEPS for 20141129_2d. The opposite is found at station n°5 for 20130305_2d and 20141129_2d. The 80% confidence interval of the PILB-HEPS simulations encompasses the observed discharge only for the 20141128_2d run at station n°2 and for the 20141129_2d run at stations n°2 and 3 (Figure 16). Given those results, it seems that there are no substantial differences

between the both HEPS strategies on these test cases.

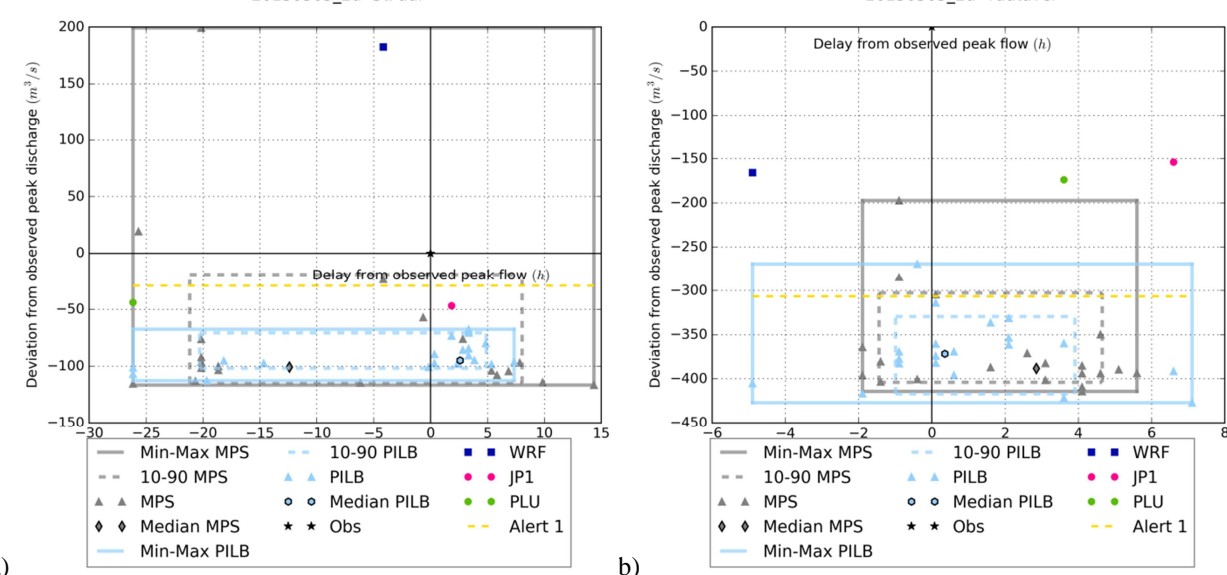

**Figure 14: Peak flow analysis at stations n°2 a) and n°5 b) for 20130305_2d. X-axis shows the delay from the observed peak time, y-axis shows the deviation from the observed peak discharge. The triangles shows the deviation of the simulations with ensemble members forcing (grey for MPS, light blue for PILB), the shapes with black contour shows the deviation of the median of the HEPS simulations with ensemble members forcing, the pink circle shows the deviation of the simulation with JP1 forcing, the green circle the deviation of the simulation with PLU forcing and the dark blue square the deviation of the simulation with deterministic WRF forcing. Alert 1 (yellow dashed line) is the warning threshold, the black star is the observation used as normalized reference.**

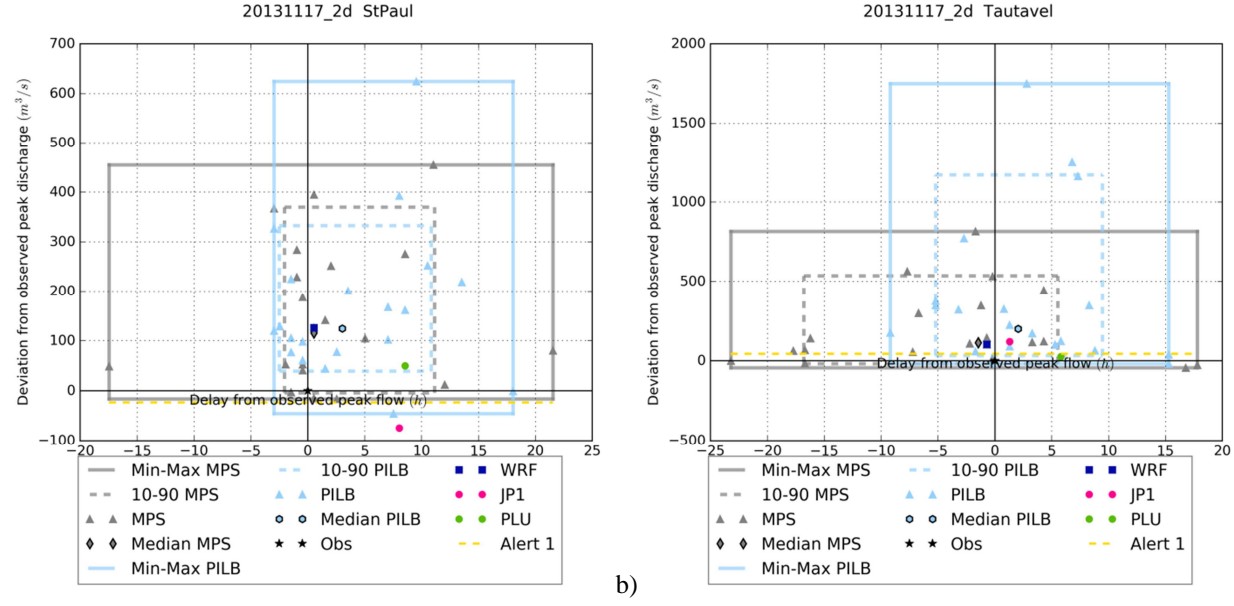

**Figure 15: Peak flow analysis at stations n°2 a) and n°5 b) for 20131117_2d. See Figure 14 for the details of the legend.**

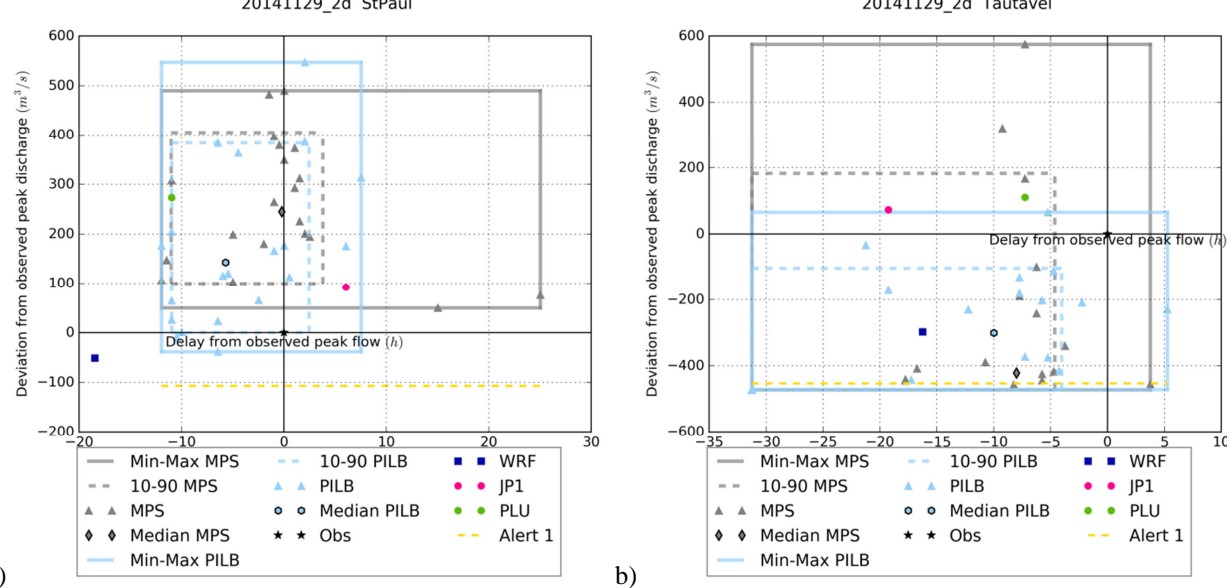

**Figure 16: Peak flow analysis at stations n°2 a) and n°5 b) for 20141129_2d. See Figure 14 for the details of the legend.**

### 5.3 System reliability for flood warning

Results of all the performed hydro-meteorological simulations lead to the conclusion that it is very difficult to correctly reproduce the spatial variability of the catchment behaviour, even forcing the hydrological model with observed rainfall. Next step was therefore to test the ability of the hydro-meteorological modelling strategies for issuing reliable flood warnings.

Let's consider a forecast event that either occurs or does not occur. For flood forecasting, it usually consists in an alert threshold exceedance. The performance of a hydro-meteorological prediction chain can be examined using a contingency table (Table 6).

|  |  | Threshold exceeded observed | |
| --- | --- | --- | --- |
|  |  | Yes | No |
| Threshold exceeded forecast | Yes | Hits (h) | False alarms (f) |
|  | No | Misses (m) | Correct negatives (n) |

**Table 6: Two-by-two contingency table for flood warning evaluation.**

Several metrics for the evaluation of flood warning performance can be derived from the contingency table by considering the number of hits (h), misses (m), false alarms (f) and corrects negatives (n) for all the simulations. The proportion correct (PC), probability of detection (POD), false alarm ratio (FAR), critical success index (CSI) and BIAS have the following properties (Nurmi, 2003):

- The PC score corresponds to the ratio of correct warning forecasts and total forecasts. PC ranges from 0 to 1, the latter being the perfect score. Note that the PC index doesn't differentiate between misses and false alarms.

- The probability of detection is the ratio of correctly forecast threshold exceedances to the total number of threshold exceeded observed. POD ranges from 0 (no hit) to 1, 1 being the best. Note that for values equal to one, there are no misses and all occurrences of the event were correctly forecast. However, POD doesn't penalize false alarms and it can be artificially improved by overforecasting.

- The false alarm ratio is the ratio of the number of false alarms to the total number of threshold exceeded forecasts. FAR ranges from 0 to 1, 0 being perfect. That is, there are no false alarms and all warning forecasts were correct. Note that FAR doesn't penalize misses and it can be artificially improved by underforecasting.

- Neither POD nor FAR can give a complete picture of forecasting success. The Critical Success Index combines both aspects of probability of detection and false alarm ratio. Therefore, CSI is more balanced and better quantifies the correspondence between the observed and forecasted occurrences. This index is sensitive to hits and penalizes both misses and false alarms. CSI values range from 0 (no hit) to 1 (no misses, no false alarms), 1 being the best. CSI ignores correct negatives as what it is expected in the forecast is to be effective in case of alert.

- The frequency bias compares the number of times an event was forecast to the number of times an event was observed. If $BIAS = 1$, both frequencies are equal and the forecast is unbiased. If $BIAS > 1 \, (< 1)$, there is an overforecast (underforecast) tendency: the event was forecast more (less) than it was observed.

As a first step, the probability of exceeding the warning threshold has been calculated for each ensemble strategy. The warning threshold that is used here is the first level alert from the flood warning center in France (SCHAPI) as plotted on Figure 12. Results are very similar for MPS-HEPS and PILB-HEPS: overall, with respect to the deterministic simulations, both ensemble strategies improve the forecast of threshold exceedance for station n°5 (Tautavel) and degrade it for station n°2 (StPaul) whereas there is no clear trend for station n°1 (Ansignan). As it has been stated in §3.2, when the hydrologic simulations are suitable for the eastern Agly (station n°2), the discharge is overestimated over the western part (station n°5). As most members of the PILB and MPS ensembles exhibit underestimations for the 04-05/03/2013 and 28-29/11/2014 events, both MPS-HEPS and PILB-HEPS result in less false alarm for station n°5 and more misses for station n°2. PILB and MPS ensembles also exhibit overestimations for the 16-18/11/2013 event but less than the deterministic simulation, results are therefore the same as for the 2 other events.

Figure 17 to Figure 19 show the results for FAR, CSI and BIAS scores at the five hydrometric sections. These scores are calculated with respect to the observed discharges and by using all the runs of the different episodes. As 48-h simulations have been performed, these scores are based on the following 7 experiments described in §5.2: 20130304_2d, 20130305_2d, 20131116_2d, 2013117_2d, 20131118_2d, 20141128_2d, 20141129_2d. Some tendencies can be highlighted from these results:

- The MPS-HEPS strategy overall performs better than the PILB-HEPS approach for the tested scores. However, both ensemble strategies scores are very similar.

- No ensemble strategy performs best for station n°2 for FAR and CSI: there is no false alarm at this station (Figure 17) and therefore, the CSI score is the best with respect to the other stations (Figure 18).

- Although the ensemble improves the peak timing in some events, it doesn't improve the issuance of warning at least according to the five tested scores: the deterministic WRF simulation always has better scores than the median of both MPS-HEPS and PILB-HEPS, except for BIAS, and sometimes better than the maximum.

BIAS shows that both ensemble strategies tend to underestimate the discharge at all the gauge-stations except station n°1, in the mountainous part of the catchment (Figure 19). That is, MPS-HEPS and PILB-HEPS tend to underestimate the discharge at all the stations except over the mountainous part of the catchment. This is an indication of the paramount importance of the orography when controlling the location of deep convection in the meteorological simulations. When orography does not play such an important role, forecasting the small-scale atmospheric features linked to the triggering and development of highly localised convective precipitation cores is more uncertain. As mentioned before, PILB-HEPS and MPS-HEPS tend to exhibit underestimations for both 20130305_2d and 20141129_2d simulations, and overestimations for the 20131117_2d run. Conversely, the observed forcing and the deterministic forecast tend to overestimate the discharge except for the two eastern stations n°4 and n°5. We find here the consequences of the hydrological model calibration: when the simulated hydrographs are suitable for the eastern Agly, the discharge is overestimated over the western part (§3.2).

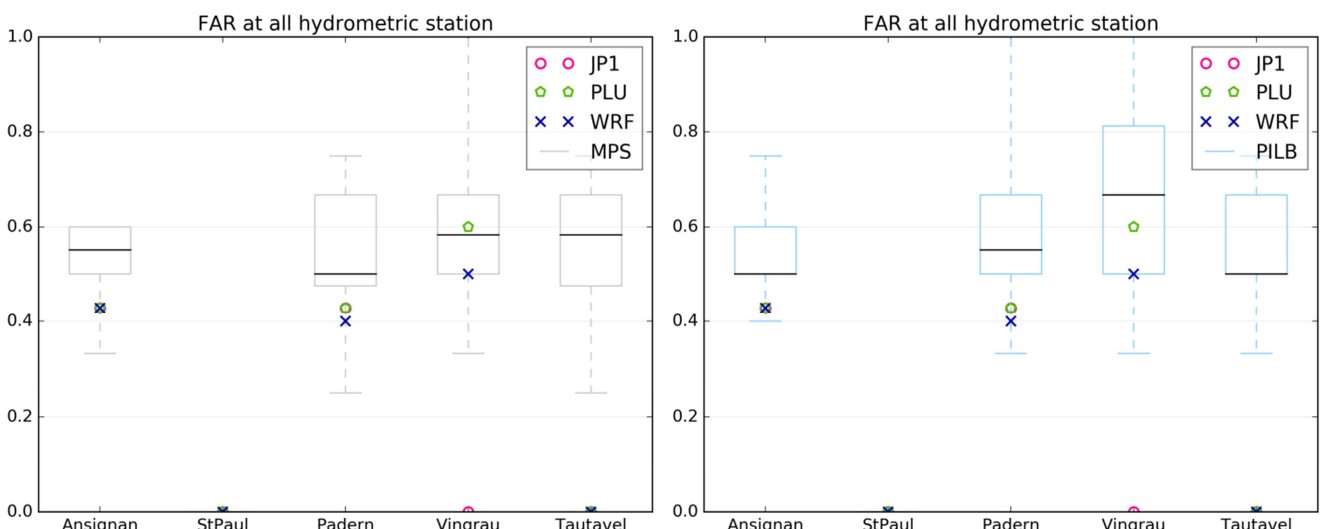

**Figure 17: False alarm ratio (FAR) scores at the five gauging stations for the 7 simulations. Statistical indices have been computed by using the observed discharge. Experiments are labelled as WRF: simulated discharge with deterministic WRF forcing, PLU: simulated discharge with PLU forcing, JP1: simulated discharge with JP1 forcing, MPS and PILB: ensemble strategies. The boxplot presents five sample statistics: the minimum, the lower quartile, the median, the upper quartile and the maximum.**

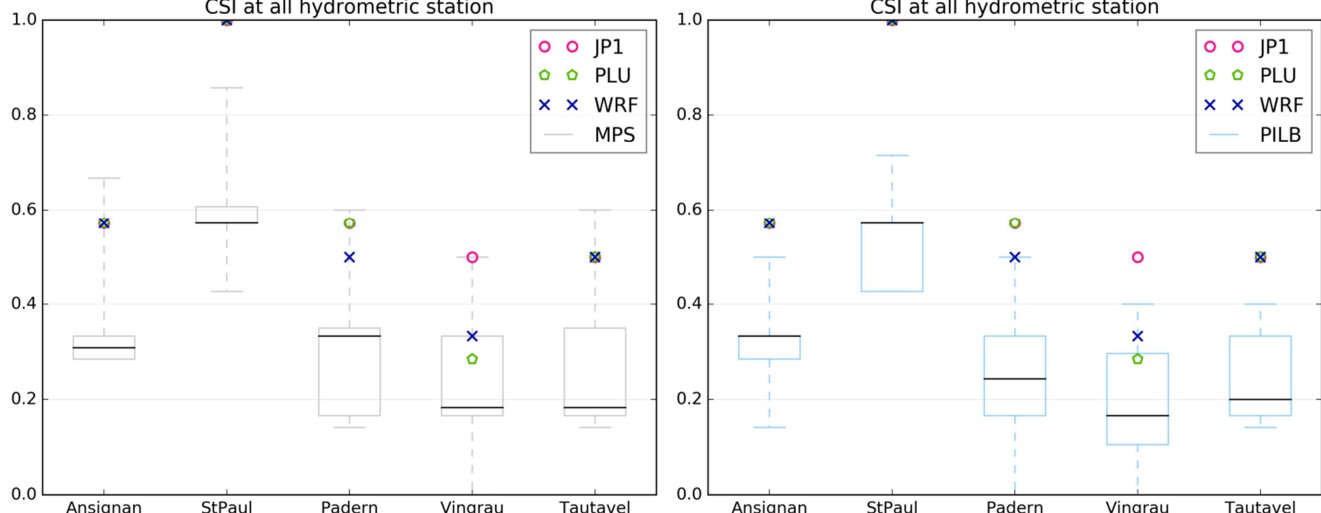

**Figure 18: as Figure 17, but for CSI.**

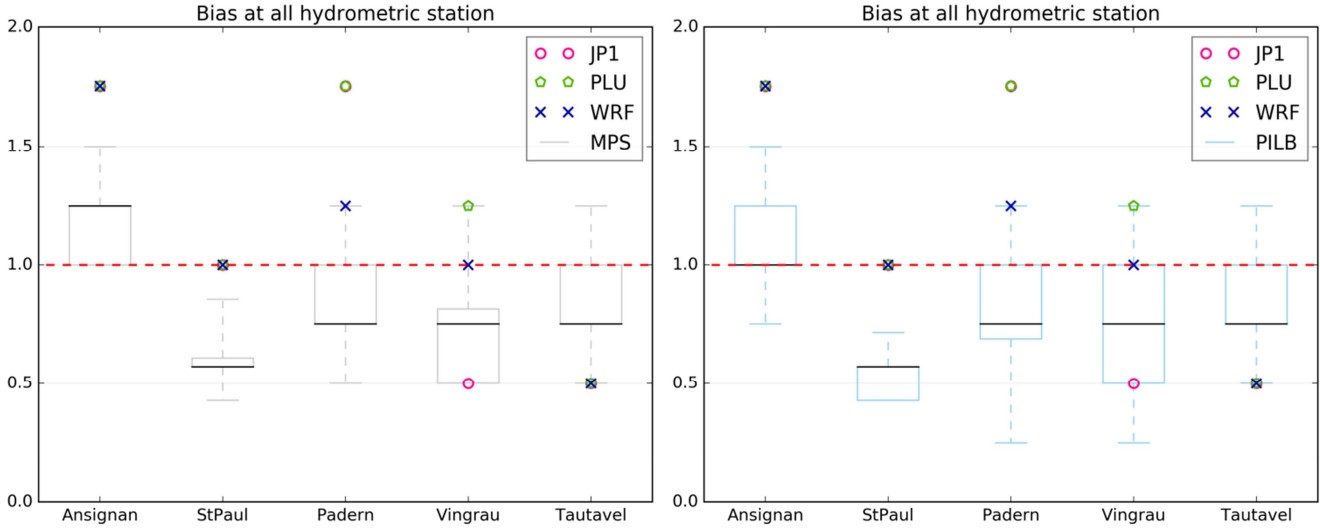

**Figure 19: as Figure 17, but for BIAS.**

Quantitative discharge forecasts can be evaluated against observed discharges but also against simulated discharges using observed forcings. As stated by several authors (Verkade et al., 2013; Bellier et al., 2017), the errors due to the parameters and structure of the hydrologic model are therefore not taken into account in the last case. This approach separates the impact of the external-scale uncertainties from these emerging from the hydrological model. Evaluations have been again performed by using the simulated discharges with observed forcing PLU and JP1 as the baseline instead of the observed
flows.

As expected, when only external-scale uncertainties are taken into account, the scores for the evaluation against simulated discharges with PLU or JP1 improve: PC, POD and CSI are higher and there are no false alarms at three stations (n°1, n°2 and n°3). However, the BIAS score shows that both ensemble strategies tend to highly underestimate the simulated discharge at all the stations, except at station n°5 when compared to PLU and at stations n°4 and n°5 when compared to JP1 (Figure 20). These stream-gauges are located over the eastern part of the catchment. Again, the deterministic WRF simulations have better scores than the median of both HEPS, except for the station n°4 and the PC, POD, FAR and BIAS scores when compared to JP1.

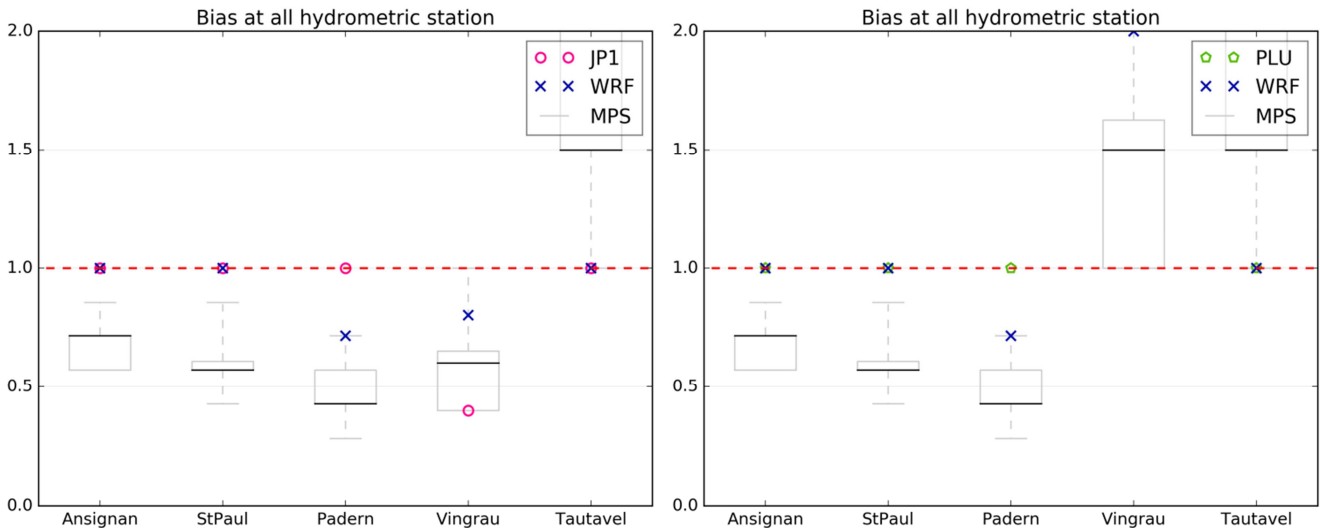

**Figure 20: Bias scores with respect to the simulated discharges with forcing PLU (left) and forcing JP1 (right) at the five gaging stations for all the simulations of the 7 simulations, WRF: simulated discharge with deterministic WRF forcing, PLU: simulated discharge with PLU forcing, JP1: simulated discharge with JP1 forcing, MPS ensemble strategies. The boxplot presents five sample statistics: the minimum, the lower quartile, the median, the upper quartile and the maximum.**

### 5.4    Overall view of the modelling performance

Binary events highlight one aspect of the forecast, especially relevant to avoid casualties, damages or economic losses (Hersbach, 2000). To obtain a more general quantification of the ensemble performances, other criteria are necessary. Here, the overall discharge forecast at the 5 gaging stations is studied by using the Continuous Rank Probability Score ($CRPS$; Matheson and Winkler, 1976). The $CRPS$ measures the differences between the forecast, $P(x)$, and observation, $P_a(x)$, expressed as cumulative distributions of one parameter $x$ (Eq. 4). This score has the dimension of the parameter and is equal to the mean absolute error (MAE) for a deterministic forecast. The following description is mainly retrieved from Hersbach (2000):

$$CRPS = \int_{-\infty}^{+\infty} [P(x) - P_a(x)]^2 dx ,\qquad\qquad(4)$$

where $x$ is the parameter of interest, herein the discharge, and $x_a$ is the value that actually occurred. $P(x)$ and $P_a(x)$ are the cumulative distributions of $x$ and $x_a$, respectively (Eqs. 5 and 6).

$$P(x) = \int_{-\infty}^{x} \rho(y)dy \, , \tag{5}$$

where $\rho(x)$ is the probability density function of the forecast $x$.

$$5 \quad P_a(x) = \mathcal{H}(x - x_a) = \begin{cases} 0 \text{ for } x < x_a \\ 1 \text{ for } x \geq x_a \end{cases} , \tag{6}$$

where $\mathcal{H}$ is the Heaviside function. The minimum value of the $CRPS$ is zero for a perfect deterministic forecast (i.e. $P(x) = P_a(x)$).

Herein, the $CRPS$ is averaged over the ensemble members and is therefore noted $\overline{CRPS}$, while the $x$ parameter corresponds to the discharge at the 5 gaging stations. The $\overline{CRPS}$ is very small for the simulations corresponding to the episode of November 2013 (i.e. 20131116_2d, 20131117_2d and 20131118_2d). This score is always below $10 \text{ m}^3 s^{-1}$ for all stations and the MPS-HEPS and PILB-HEPS strategies. Conversely, the $\overline{CRPS}$ is quite high – above $50 \text{ m}^3 s^{-1}$ – for the numerical runs of the event of November 2014 (i.e. 20141128_2d and 20141129_2d), especially at the station n°5. That is, the cumulative distributions of discharge are similar between the HEPSs and the observed discharges for the event of November 2013, but they are dissimilar for the episode of November 2014. Concerning the experiments for the episode of March 2013 (i.e. 20130304_2d and 20130305_2d), the $\overline{CRPS}$ is low for stations n°1 and n°3 (below $15 \text{ m}^3 s^{-1}$) and higher for stations n°2, n°4 and n°5 (close to or above $20 \text{ m}^3 s^{-1}$).

To evaluate more easily the performances of the ensemble strategies, their performances are also compared against the efficiency of a reference forecast by using the skill score with respect to the $\overline{CRPS}$ (Eq. 7) (Bontron, 2004):

$$CRPSS = 1 - \frac{\overline{CRPS}}{CRPS_{ref}} \, , \tag{7}$$

The chosen reference forecast is the simulation performed with the deterministic forecast (WRF) and in that case the $\overline{CRPS}$ skill score writes as follows:

$$CRPSS = 1 - \frac{\overline{CRPS}}{MAE(WRF)} \, , \tag{8}$$

A $CRPSS$ of 1 corresponds to a perfect forecast ($\overline{CRPS} = 0$), while a value of 0 indicates that the HEPS and the reference forecast have the same performances ($\overline{CRPS} = MAE(WRF)$). Negative skill scores denote that the reference prediction performs better than the HEPS ($\overline{CRPS} > MAE(WRF)$).

Figure 21 shows that the two ensemble strategies exhibit very similar skill score $CRPSS$:

- In general, both ensemble strategies perform better than the deterministic WRF experiment, except for 20130304_2d and 20130305_2d.
- The main differences between both ensemble strategies are found for the 20131118_2d experiment: PILB clearly outperforms MPS at all the stream-stations

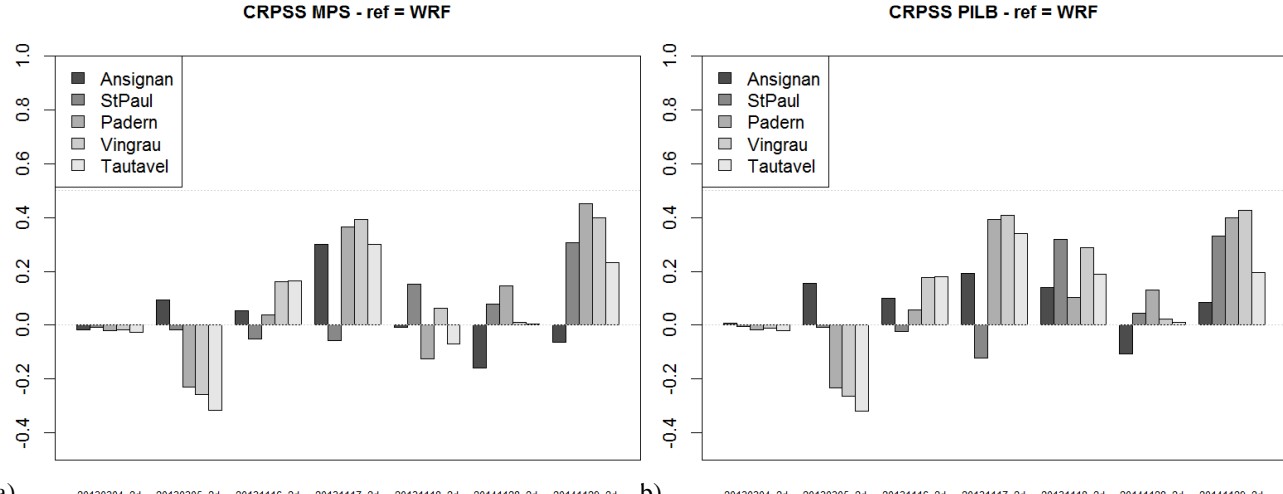

**Figure 21:** $\overline{CRPS}$ **skill scores of the seven 48-h experiments and at the 5 hydrometric stations for the: (a) MPS-HEPS and (b) PILB-HEPS strategies. Reference forecast is the deterministic WRF experiment.**

As stated before, selecting the runoff simulation driven by the deterministic weather forecast as the reference does not account for the errors due to the hydrological model. The $\overline{CRPS}$ skill score can also be calculated by using the simulation performed with the observed precipitation fields (PLU and JP1) as the reference:

$$CRPSS_{PLU} = 1 - \frac{\overline{CRPS}}{MAE(PLU)}$$
$$CRPSS_{JP1} = 1 - \frac{\overline{CRPS}}{MAE(JP1)}$$

(9)

Not surprisingly, both ensemble strategies have an overall lower performance when compared with the PLU and JP1 driven runoff simulations, except for event of November 2013. It is interesting to notice that for the 20131118_2d run, the PILB driven runoff forecasts outperform the radar driven discharge simulation (Figure 22, right). This is consistent with the previous analyses: events with relatively moderate peak discharge – as the event of November 2013– are not correctly simulated by MARINE whatever the observed forcing (Table 4), whereas the $\overline{CRPS}$ is very low for the ensemble simulations of the event of November 2013. As stated before, a low $\overline{CRPS}$ means that the cumulative distributions of discharge are similar between both HEPSs and the observed discharges for the event of November 2013, but they are dissimilar between the simulations with both observed forcings and observed discharges for the same event. This may be related to the fact that

MPS-HEPS and PILB-HEPS exhibit overestimations for this event maybe compensating errors in the model structure that prevent the simulation with observed forcings for this event to be efficient. Both ensemble strategies outperform the hydrological simulations driven by observed forcings (PLU and JP1) for the mountainous station (n°1: Ansignan) and the 20141128_2d, 20141129_2d, 20131116_2d and 20131118_2d runs. This result is consistent with the difficulty to obtain satisfactory observations of rainfall in mountainous areas owing to sparse rain-gauge deployment and beam radar blockage.

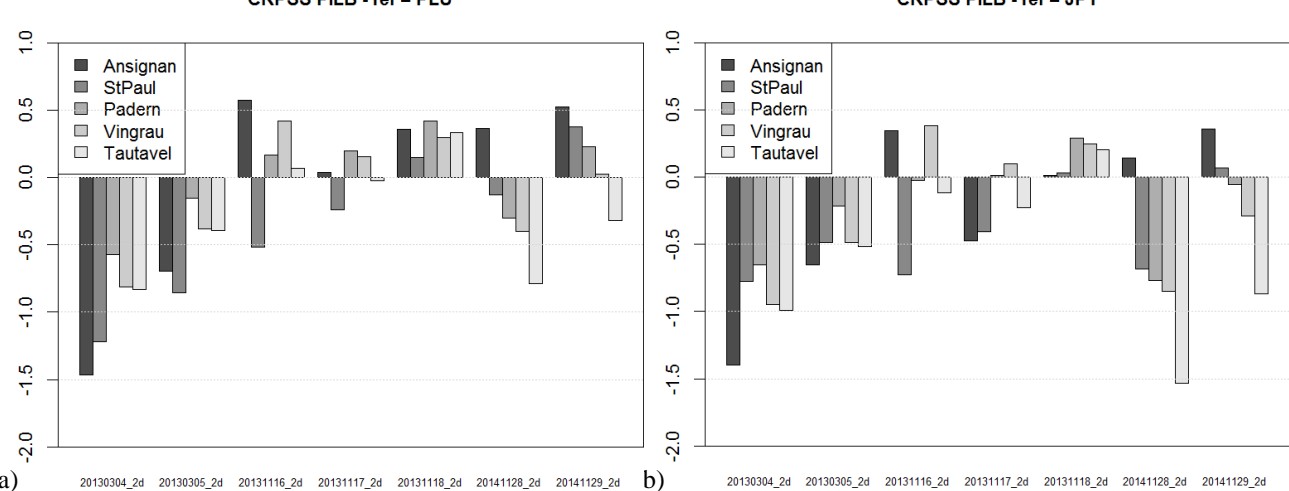

a)   b)

**Figure 22: As Fig. 21, but just for the PILB-HEPS and the (a) PLU and (b) JP1 as reference.**

## 6   Conclusion

One of the main scientific aims of the HyMeX program is to improve the hydro-meteorological forecasting of flash floods over the Western Mediterranean region. To this end, three of the most important floods that recently developed over the Agly basin have been selected as study cases. Flood forecasting is a challenging task over this region: high spatial and temporal variability in convective cores and rainfall intensity, strong nonlinearities in the rainfall-runoff transformation and antecedent moisture conditions lead to a myriad of hydrological responses. This work has focussed in coping with uncertainties emerging from the initial and lateral boundary conditions and formulation of numerical weather prediction models. To this end, potentialities of MPS-HEPS and PILB-HEPS ensembles have been examined so as to produce suitable flood forecasts over the Agly basin. Main conclusions are:

- A better ensemble generation strategy at regional scale has not been found. Similarities in the performance of the MPS and PILB approaches indicate that both sources of external-scale uncertainty contribute similarly to produce adequate levels of skill and spread in the probabilistic quantitative precipitation forecasts.

- Ensemble hydro-meteorological simulations have resulted satisfactory for alarm detection, even if individual ensemble members can be far from the observations. Alarm systems benefit from large hydro-meteorological ensemble spreads.

- The overall HEPS performances improved the deterministic driven runoff simulations.

Some unexpected results also rise interesting questions. For instance, the November 2013 event was poorly simulated using both observed forcings, but ensemble strategies improved the overall discharge forecast. What is the specificity of the November 2013 event that makes it poorly simulated? Is it due to the radar and rain-gauges location? Or to the initial state of the catchment? Is it due to the model structure itself that does not represent all the hydrological processes involved (karstic system and snowmelt mainly)? These issues require further investigations and probably more test cases. The next logical approach will be to estimate the uncertainties in the hydrological modelling. Performing hydrological model ensemble to test the errors due to the model calibration is time consuming. However, according to Douinot et al. (2017), it is also useful in identifying the strengths and weaknesses of the model when simulating the distinct hydrological processes. Hopefully, the future implementation of an hydrological model ensemble will provide the beginning of the answers to the above questions.

**Acknowledgements**

This work was carried out in the framework of the PGRI-EPM project (Prévision et gestion du risque d'inondation en l'Eurorégion Pyrénées Méditerranée) funded by the call for projects "Développement durable, Ressource eau – Gestion des risques" of the Eurorégion Pyrénées-Méditerranée. This work has also been sponsored by several Spanish research projects (PCIN-2015-221 (METEOforSIM) and CGL2017-82868-R (COASTEPS), which are partially supported with FEDER funds) and by the French Central Service for Flood Forecasting (SCHAPI). It is a contribution to the HyMeX programme. The authors would like to thank Béatrice Vincendon (CNRM-Météo France) for providing the ANTILOPE radar reanalysis and the regional flood forecasting service, the Service de Prévision des Crues Méditerranée Ouest (SPCMO) for providing the rain-gauge data.

**Code/Data availability**

The WRF model is free. Readers can find the code in the web page of NCAR-UCAR (National Centre for Atmospheric Research - Mesoscale and Microscale Meteorology Laboratory https://www2.mmm.ucar.edu/wrf/users/download/get_sources.html). The MARINE model is governed by the CeCILL license under French law (http://www.cecill.info) and can be accessed by contacting Hélène Roux (helene.roux@imft.fr). Research data are available under request.

**Author contribution**

HR, AA and RR provide theoretical background, design the methodology and analyse data. AA run the meteorological model and write the corresponding part. HR run the hydrological model and write the corresponding part. All the authors participate in the discussions and review the manuscript.

**Competing interests**

The authors declare that they have no conflict of interest.

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
