# Peer review of "Evaluation of two hydro-meteorological ensemble strategies for flash flood forecasting over a catchment of the eastern Pyrenees"

_Natural Hazards and Earth System Sciences, 2019_

## Referee Comment (RC1) · Anonymous Referee #1 · 11 Oct 2019

The paper presents a comparison of two ensemble strategies to forecast flash floods in the eastern Pyrenees. This topic is not totally new, but the problem is still an unsolved problem of the hydro-meteorological modelling and is worth further investigation as that one undertaken by authors of this paper. I found the paper interesting and well written. I recommend it for publication after minor revision. Comments: Fig. 1 I find it difficult to understand. It's not easy to distinguish position of radar, discharge gaging stations and the dam. I suggest to you use more distinguishable marks and associated legend to show relevant elements. P. 4 L 2. Area of tributaries and basin intercepted by dam are mentioned. I suggest to give information about total basin area that is reported in Table 1 (is it 1053 km2?). I think this is relevant to understand how dam

can affect hydrograph of outlet section. P. 4 L. 18. Rain-gauge network are provided by the regional flood forecast service. I suggest to add a reference to figure 2 that shows locations of raingages across the investigated area. P.7 L2-3 There's no need to give details about how Thiessen polygon interpolation method works. P. 8 what is the spatial resolution used in the hydrological model? P.8 L22. How the spatial daily root-zone humidity maps are used to initialize the hydrological model? P9 L16 The Bransby Williams formula is used for computing time of concentration. There are many equations in literature for time of concentration and the spread they do is very high. Why did you choose this one? May the model performance assessment affected by the choice of formula for time of concentration? P 9 L 26 Can raingages distribution explain, at least partially, the different performance of the model across the basins?

---

## Referee Comment (RC2) · Anonymous Referee #2 · 11 Nov 2019

This article presents the creation of two meteorological NWP ensemble forecasts, 1) by perturbing the initial and lateral boundary conditions, 2) different physical parameterisations, and then forcing these through a hydrological model of a Pyreneean catchment. The forecasts are assessed for three different flood events against in-situ observations. An assessment of the systems' ability to provide reliable flood warnings against observed flood threshold exceedances is also performed. Overall both ensemble strategies provided similar performances and were able to provide more reliable flood warnings than a deterministic strategy.

This article provides an important contribution to the topic of ensemble forecasting for

flood hazards, therefore it is within the scope of this journal. I believe this manuscript is acceptable for publication after minor revisions are made. One general comment is whether it would be possible to reduce the number of figures in the manuscript as 22 is quite a lot. For example in section 5.1 there are 6 figures, but I believe that it is only necessary to retain figures 10 and 11 as these contain the most important information regarding the verification of the SREPS. I would also like the authors to be more explicit about why these two particular ensemble strategies were chosen, what differences may be expected from them and why these differences were not observed. The choice of hydrological model also needs further justification given its omission of karstic streamflow contributions which could prove important within the study catchment.

Further to the above comments, please could the authors also address the following points:

1. Page 1 line 30: replace 'large sea surface temperature' with 'high sea surface temperature' 2. Page 3 line 8: replace 'its' a real challenge' with 'it is challenging' 3. Figure 2: Could the dots and stars in 2b be made larger and also be surrounded by a white halo. It would also be useful if the black text could also have a halo 4. Page 5 line 1: Could the authors provide a little more explanation behind the runoff coefficient being greater than 1 in the Tautavel catchment for the first event. Table 3 seems to suggest that the soil moisture is similar for all three events. If there was a supply from the karstic system wouldn't this influence all three events as well as other catchments? I wonder if this could be related to the amount of snowmelt or snowfall since the event in question occurred in March, could the authors comment on this? 5. Page 7 Figure 2: I can't read the grey labels for the rain gauge names, could these be enlarged and also maybe with a white halo? 6. Page 8 Section 3.1: Given the previous discussion about the possible role of contributions from karstic streams, it concerns me that the hydrological model used in this study does not account for this process. Could the authors comment on the significance of karstic streamflow contributions in this catchment and the possible consequences of its exclusion from the hydrological model upon streamflow accuracy?

7. Page 10 Table 4: It seems like the event of 20131116 has a very low efficiency in all but one station which is located at the upper end of the catchment. In their analysis the authors suggest that this is because events with a moderate peak discharge are not well simulated by MARINE. Why is this the case, is it due to the routing scheme in MARINE? From these poor scores I think this event should be eliminated from the rest of the analysis in the manuscript, could the authors comment on this? 8. Page 12 Line 6: Has 'MPS' being defined previously in the manuscript? If not could the full definition be given? 9. Page 13 Line 6: Please give the definition for the IC and LBC acronyms 10. Page 13 Line 26: How do the different microphysical and PBL schemes add up to 20 ensemble members? 11. Page 14 Line 8: Define the CCN acronym 12. Page 14 Line 23: Add the word 'catchment' so the sentence reads '...a single medium-sized catchment is a challenging...' 13. Page 15 Figure 6: Add the following column titles: JP1, MPS, PILB. The same for figures 7 and 8. However I think these figures could all be removed from the manuscript and maybe put in supplementary material in order to cut down the number of figures in the manuscript. 14. Page 18 Figure 9: What is the CTRL referring to? In the caption replace 'the best and worst ensemble members' with 'the tails of the ensemble' 15. Page 20 Line 8: Are the 7735 grid points just within the catchment or is this over a wider area? 16. Page 21 Figure 11: Could a title and units be added to the legends 17. Page 24 Figure 14, 15, 16: I find it hard to see the grey boxes, could these be made a bit darker and maybe thicker so that they stand out more? 18. Page 26 Line 18: What is the warning threshold that is used? 19. Page 27 Line 6: Replace 'excepted' with 'except' 20. Page 27 Figure 17, 18, 19: I'm unclear what the two separate graphs in each figure show, could the authors improve the titles and/or captions? 21. Page 28 Line 5: Define QDF if not already defined 22. Page 29 Line 2: Replace 'excepted' with 'except', also occurs on page 30 line 24 23. Page 32 Line 12: Could the authors provide more discussion about why there was little difference between the two ensemble strategies? Why were these two different strategies chosen, what differences may have been expected and why do they think these differences weren't observed?

---

## Author Comment (AC1) · 19 Dec 2019

We would like to thank the Referee #1 for his evaluation. Please find below the point-by-point replies for the comments of Anonymous Referee # 1 (The reviewer's comments are in italic).

*Comments by Anonymous Referee #1:*

*Fig. 1 I find it difficult to understand. It's not easy to distinguish position of radar, discharge gaging stations and the dam. I suggest to you use more distinguishable marks and associated legend to show relevant elements.*

We followed your suggestion and we have introduced more distinguishable marks. The legend is detailed in the caption. The reviewed figure also includes the recommendations from Referee #2:

[Figure]

**Figure 1: a) Location of the Agly catchment. The pink star illustrates the position of the meteorological radar while shaded grey areas denote the karstic areas underlying the Agly catchment (from BDLISA v.2: Base de Donnée des Limites des Systèmes Aquifères, https://bdlisa.eaufrance.fr/ accessed June 18, 2019). b) Digital terrain model of the Agly catchment (Source: IGN; MNT BDALTI). Also included the main tributaries (blue lines, source: IGN, BD CARTHAGE), the radar location (pink star: OPOUL RADAR), the discharge gaging stations (black dots), the dam (white square) and the outlet (white circle).**

*P. 4 L 2. Area of tributaries and basin intercepted by dam are mentioned. I suggest to give information about total basin area that is reported in Table 1 (is it 1053 km2?). I think this is relevant to understand how dam can affect hydrograph of outlet section.*

The total basin area of 1053 km$^2$ is reported on Table 1 for information purpose only as the gauging stations studied in the paper are not impacted by the dam: n°1 Ansignan and n°2 St-Paul-de-Fenouillet are located upstream the dam and n°3 Padern, n°4 Vingrau and n°5 Tautavel are located on a tributary of the Agly river.

*P. 4 L. 18. Rain-gauge network are provided by the regional flood forecast service. I suggest to add a reference to figure 2 that shows locations of raingages across the investigated area.*

Reference added:

> **Figure 2: Spatial variability of the cumulative rainfall for event 20130304_3d (top), 20131116_4d (middle) and 20141128_4d (bottom), according to the observations: PLU (left) the operational hourly rain-gauge network (from Hydroreel, Serveur de données hydrométriques en temps réel, Bassin Rhône-Méditerranée et Région Auvergne-Rhône-Alpes, https://www.rdbrmc.com/hydroreel2/listestation.php accessed on November 20,2019) and JP1 (right) 1 km$^2$ merging of radar data and rain-gauges measurements.**

*P.7 L2-3 There's no need to give details about how Thiessen polygon interpolation method works.*

The explanations about the Thiessen polygon interpolation method have been removed.

*P. 8 what is the spatial resolution used in the hydrological model?*

The spatial resolution of the MARINE model on the Agly subcatchment is Δx=Δy=500 m. It has been added in the text (§ 3.2):

> "The spatial resolution of the MARINE model on all the Agly subcatchments is of 500 m."

*P.8 L22. How the spatial daily root-zone humidity maps are used to initialize the hydrological model?*

To initialize the hydrological model, we use the output Météo-France's SIM operational chain corresponding to a saturation state, that is, the ratio of the soil water content to the soil storage capacity. The initial soil water content is therefore directly obtained by multiplying the saturation state by the soil storage capacity of each cell. This has been clarified in the text (§ 3.1):

> "This is done by using the spatial daily root-zone saturation state, i.e. the ratio of the soil water content to the soil storage capacity at a spatial resolution of 8×8 km, output from Météo-France's SIM operational chain (Habets et al., 2008). The initial soil water content for MARINE is therefore directly obtained by multiplying the saturation state by the soil storage capacity of each cell."

*P9 L16 The Bransby Williams formula is used for computing time of concentration. There are many equations in literature for time of concentration and the spread they do is very high. Why did you choose this one? May the model performance assessment affected by the choice of formula for time of concentration?*

This formula has been adopted as it performed reasonably well when compared with characteristic minimum times of rise of observed hydrographs for Mediterranean catchments. However the point there was mostly to normalize the peak time delay (P9 L11 in equation 1) with a characteristic time of the catchment so the most important point is to always use the same procedure to make this term dimensionless in the cost function of equation 1. This has been clarified in the text (§ 3.2):

> "Here, the formula for time of concentration is only used to normalize the peak time delay in the third term of equation 1 with a characteristic time of the catchment, so the most important point is to always use the same procedure to make this term dimensionless."

*P 9 L 26 Can raingages distribution explain, at least partially, the different performance of the model across the basins?*

It is difficult to link directly the rain-gauges distribution with the performances of the model for 2 reasons:

- First, the rain-gauge network is quite dense in this catchment and rather well distributed: with 19 rain-gauges for an area of around 1000 $km^2$, the rain-gauges density is about 1 for 50 $km^2$ whereas the rain-gauge density for the full network over mainland France is of 1 for 120 $km^2$ (Mounier et al., 2012).

- Secondly, it's not always for the same part of the catchment that the model has the best performance: it depends on the event. Therefore, the same distribution of rain-gauges sometimes leads to a correct simulation in term of $L_{NP}$ cost function (equation 1 in the manuscript) for a given even, while leads to an unsatisfactory simulation for another event.

This has been added in the manuscript (§ 3.2):

> "This result doesn't seem to be directly linked with the rain-gauged distribution because first of all, the rain-gauge network is quite dense in this catchment and rather well distributed: with 19 rain-gauges for an area of around 1000 $km^2$, the rain-gauges density is about 1 for 50 $km^2$ whereas the rain-gauge density for the full network over mainland France is of 1 for 120 $km^2$ (Mounier et al., 2012). In addition, it's not always for the same part of the catchment that the model has the best performance: it depends on the event. Therefore, the same distribution of rain-gauges sometimes leads to a correct simulation in term of $L_{NP}$ cost function (Eq. 1) for a given even, while leads to an unsatisfactory simulation for another event."

Mounier, F., Lassègues, P., Gibelin, A.-L., Céron, J.-P. and Veysseire, J.-M.: Radar-guided control and interpolation of rain gauge precipitation data over France. Report EURO4M project (European Reanalysis and Observations for Monitoring project). http://www.euro4m.eu/Publications/Report_Flore_Mounier_EURO4M_201203.pdf, accessed December 6, 2019. 2012.

---

## Author Comment (AC2) · 19 Dec 2019

We would like to thank the Referee #2 for his evaluation. Please find below the point-by-point replies for the comments of Anonymous Referee # 2 (The reviewer's comments are in italic).

*Comments by Anonymous Referee #2:*

*One general comment is whether it would be possible to reduce the number of figures in the manuscript as 22 is quite a lot. For example in section 5.1 there are 6 figures, but I believe that it is only necessary to retain figures 10 and 11 as these contain the most important information regarding the verification of the SREPS.*

We would prefer to keep these figures in the manuscript as we think they are relevant for a better understanding of the whole study.

*I would also like the authors to be more explicit about why these two particular ensemble strategies were chosen, what differences may be expected from them and why these differences were not observed.*

When forecasting deep moist convection and heavy rainfall with high-resolution numerical weather prediction models, the outputs are mainly impacted by two sources of errors. One source is the inaccuracies present in the exact representation of the initial and lateral boundary conditions (IC/LBCs). The other source is due to imperfect representation of physical processes via parameterizations. Nowadays, atmospheric ensembles are built to cope with both kinds of distinct uncertainties by perturbing the IC/LBCs or by considering multiple combinations of well-tested numerical schemes. The most appropriate methods for generating Hydrological Ensemble Predictions Systems (HEPS) is a subject under continuous investigation and more methods could come in the future. Here we followed the state-of-the-art approach used in many other hydro-meteorological studies.

Even if PILB and MPS ensembles address different kinds of uncertainties, these sources of error would be expected to have a comparable impact on the skill of quantitative precipitation forecasts (QPFs) if the EPS is properly designed. This seems to be the case of our configuration. Comments on the specific purpose and value of the PILB and MPS ensemble strategies and on the method to avoid under-dispersive behaviour of PILB, have been added in sections 1 and 4.

§ 1:

"However, the most appropriate methods for generating HEPSs and the quantification of their added value are still under assessment (Cloke and Pappenberger, 2009; Cloke et al., 2013). Further efforts devoted to examine the predictive skill of both ensemble strategies and how the external-scale uncertainties spread into the HEPSs become paramount for the optimal design of hydrometeorological operational chains over the flood-prone Western Mediterranean area."

§ 4.1:

"However, perturbed IC/LBCs can produce inadequate spread in the short range, before error growth on the synoptic scale becomes non-linear (Gilmour et al., 2001). Therefore, the implemented PILB ensemble is based on dynamically downscaling these 20 ECMWF-EPS members exhibiting maximum perturbations in the initial and lateral boundaries conditions over the WRF domain."

*The choice of hydrological model also needs further justification given its omission of karstic streamflow contributions which could prove important within the study catchment.*

The hydrological model has been chosen as it represents physical processes using equations derived from classical mechanics while taking into account the spatial variability of both catchment properties and forcing inputs. Karstic areas are not explicitly represented as physically-based models for karstic streamflow contributions are usually site-specific: most of the modelling approaches for karts systems that are not site-specific are conceptual ones

(Bakalowicz, 2005). However the study doesn't focus on the performances of the hydrological model that of course could have been improved. The main purpose is the potential of ensemble strategies to improve flash flood forecasting. That's why NWP model driven runoff simulations have been compared both against the observed discharges and against the observed rain-gauge and radar precipitation driven runoff runs. As already mentioned in the manuscript, the errors due to the parameters and structure of the hydrologic model are therefore not taken into account when comparing NWP model driven runoff simulations against the observed rain-gauge and radar precipitation driven runoff runs. This approach separates the impact of the external-scale uncertainties from these emerging from the hydrological model.

Bakalowicz, M.: Karst groundwater: a challenge for new resources. Hydrogeology Journal, 13(1), 148-160, doi: 10.1007/s10040-004-0402-9, 2005.

*Further to the above comments, please could the authors also address the following points:*

*1. Page 1 line 30: replace 'large sea surface temperature' with 'high sea surface temperature'*

Done

*2. Page 3 line 8: replace 'its' a real challenge' with 'it is challenging'*

Done

*3. Figure 2: Could the dots and stars in 2b be made larger and also be surrounded by a white halo. It would also be useful if the black text could also have a halo*

We followed your suggestion and we have introduced more distinguishable marks and white halo around the text and marks. The reviewed figure also includes the recommendations from Referee #1:

[Figure]

**Figure 1: a) Location of the Agly catchment. The pink star illustrates the position of the meteorological radar while shaded grey areas denote the karstic areas underlying the Agly catchment (from BDLISA v.2: Base de Donnée des Limites des Systèmes Aquifères, https://bdlisa.eaufrance.fr/ accessed June 18, 2019). b) Digital terrain model of the Agly catchment (Source: IGN; MNT BDALTI). Also included the main tributaries (blue lines, source: IGN, BD CARTHAGE), the radar location (pink star: OPOUL RADAR), the discharge gaging stations (black dots), the dam (white square) and the outlet (white circle).**

*4. Page 5 line 1: Could the authors provide a little more explanation behind the runoff coefficient being greater than 1 in the Tautavel catchment for the first event. Table 3 seems to suggest that the soil moisture is similar for all three events. If there was a supply from the karstic system wouldn't this influence all three events as well as other catchments? I wonder if this could be related to the amount of snowmelt or snowfall since the event in question occurred in March, could the authors comment on this?*

The soil moisture at the beginning of the event is of 65% when the second highest initial soil moisture is of 58% for 20141128_4d. This is significantly different, especially knowing that the outputs of the SIM model used as

initialization for the MARINE model have a limited variation range, mainly between 30% and 70%. A supply of the karstic system can influence only one event, depending on the previous filling conditions of the karst, however it's not the most likely option as hydrogeological studies of the areas conclude that there are losses due to the karstic system in the Verdouble catchment. The amount of snowmelt has not been considered for this part of the catchment as the Corbières are quite low mountains that culminate at approximately 1000 m with usually no snowpack. However it is true that winter 2013 has been very cold and there was a snowfall episode at the very end of February 2013 over the Eastern Pyrenees and Corbières, with snow above 700 to 800 m, which continues during the day on March 1st. This has been modified in the text (§ 3.3):

> "There is no definitive explanation for that, but several possibilities can be considered: (i) the very high soil moisture at the beginning of the event (65%, Table 3) which can contribute to the runoff at the outlet via subsurface flows; (ii) an amount of snowmelt as there was a snowfall episode at the very end of February 2013 over the Eastern Pyrenees and Corbières, with snow above 700 to 800 m; (iii) the uncertainties in the discharge and precipitation measurements; (iv) a possible supply from the karstic system (Figure 1) however this possibility is pretty unlikely as hydrological studies conclude to only losses in the Verdouble catchments due to the karstic system (Ladouche et al., 2004)."

Ladouche, B., Dörfliger, N.: Evaluation des ressources en eau des corbières. Phase I – Synthèse de la caractérisation des systèmes karstiques des Corbières Orientales, Tecnical report BRGM, available online http://infoterre.brgm.fr/rapports/RP-52919-FR.pdf accessed December 06, 2019, 2004.

*5. Page 7 Figure 2: I can't read the grey labels for the rain gauge names, could these be enlarged and also maybe with a white halo?*

Done

[Figure]

**Figure 2:** Spatial variability of the cumulative rainfall for event 20130304_3d (top), 20131116_4d (middle) and 20141128_4d (bottom), according to the observations: PLU (left) the operational hourly rain-gauge network (from Hydroreel, Serveur de données hydrométriques en temps reel, Bassin Rhône-Méditerranée et Région Auvergne-Rhône-Alpes, https://www.rdbrmc.com/hydroreel2/listestation.php accessed on November 20,2019) and JP1 (right) 1 km² merging of radar data and rain-gauges measurements.

*6. Page 8 Section 3.1: Given the previous discussion about the possible role of contributions from karstic streams, it concerns me that the hydrological model used in this study does not account for this process. Could the authors comment on the significance of karstic streamflow contributions in this catchment and the possible consequences of its exclusion from the hydrological model upon streamflow accuracy?*

According to hydrogeological studies of the area, there are only losses in the Agly and Verdouble catchments due to the karstic system. These losses contribute to the streamflow of 2 resurgences draining the Corbières massif but located outside of the Agly catchment (Font-Estramar and Font-Dame resurgences) (Salvayre, 1989). The average loss rates are estimated between 0.3 and 1.5 m³/s for the Agly depending on the river discharge and between 0.7-2 m³/s on the Verdouble (Ladouche et al., 2004). These are average estimates based on observed discharges and assumptions about the functioning of the karst system and they can be considered small enough not to be explicitly represented during flash flood. These losses can however be implicitly taken into account in the hydrological model by increasing the storage capacity of the catchment during calibration. Moreover, as previously mentioned, the purpose of the study was not the performances of the hydrological model alone but the potential of ensemble strategies to improve flash flood forecasting. That's why NWP model driven runoff simulations have been compared both against the observed discharges and against the observed rain-gauge and radar precipitation driven runoff

runs. The errors due to the parameters and structure of the hydrologic model are therefore not taken into account when comparing NWP model driven runoff simulations against the observed rain-gauge and radar precipitation driven runoff runs. This approach separates the impact of the external-scale uncertainties from these emerging from the hydrological model.

A description of the karstic system contributions has been added in the text (§ 2.1):

> "According to hydrogeological studies of the area, there are only losses in the Agly and Verdouble catchments due to the karstic system. These losses contribute to the streamflow of two resurgences draining the Corbières massif but located outside of the Agly catchment (Font-Estramar and Font-Dame resurgences) (Salvayre, 1989). The average loss rates are estimated between 0.3 and 1.5 $m^3$/s for the Agly depending on the river discharge and between 0.7-2 $m^3$/s on the Verdouble (Ladouche et al., 2004). These are only average estimates based on observed discharges and assumptions about the functioning of the karst system but they can be considered small enough not to be explicitly represented in flash flood simulations."

Ladouche, B., Dörfliger, N.: Evaluation des ressources en eau des corbières. Phase I – Synthèse de la caractérisation des systèmes karstiques des Corbières Orientales, Tecnical report BRGM, available online http://infoterre.brgm.fr/rapports/RP-52919-FR.pdf accessed December 06, 2019, 2004.

Salvayre, H.: Les karsts des Pyrénées-Orientales (Caractères hydrogéologiques et spéléologiques généraux). In: Karstologia : revue de karstologie et de spéléologie physique, n°13, 1er semestre 1989. pp. 1-10; doi: https://doi.org/10.3406/karst.1989.2199, https://www.persee.fr/doc/karst_0751-7688_1989_num_13_1_2199, 1989.

*7. Page 10 Table 4: It seems like the event of 20131116 has a very low efficiency in all but one station which is located at the upper end of the catchment. In their analysis the authors suggest that this is because events with a moderate peak discharge are not well simulated by MARINE. Why is this the case, is it due to the routing scheme in MARINE? From these poor scores I think this event should be eliminated from the rest of the analysis in the manuscript, could the authors comment on this?*

Yes, the events with relatively moderate peak discharge are usually not correctly simulated by MARINE because the flow over the hillslope and in the drainage network is represented with the kinematic wave assumption valid for high flow velocity. However it is difficult to say when this assumption ceases to be valid for overland flow due to local conditions. I do not think it is necessary to withdraw events that do not produce good results because they also have lessons to learn. Here for instance, the ensemble strategies outperform the radar driven discharge simulation for the event of 20131116 which may also be indicative of questionable quantitative precipitation estimates.

*8. Page 12 Line 6: Has 'MPS' being defined previously in the manuscript? If not could the full definition be given?*

MPS stands for mixed-physics ensemble; it is already defined § 1.

*9. Page 13 Line 6: Please give the definition for the IC and LBC acronyms*

IC stands for initial condition and LBC for lateral boundaries conditions, they are defined on § 1.

*10. Page 13 Line 26: How do the different microphysical and PBL schemes add up to 20 ensemble members?*

Each possible combination of the 5 different cloud microphysical schemes with the 4 distinct PBL parameterizations is considered to build a member of the MPS ensemble. These means a total of 20 pairs microphysics-boundary layer. Corresponding sentence in section 4.2 has been rewritten to avoid any confusion:

"The MPS ensemble has been generated using all possible pairs (cloud microphysics-boundary layer) between the following schemes, summing up to 20 members:"

*11. Page 14 Line 8: Define the CCN acronym*

It is define just before the acronym: cloud condensation nuclei. Capital letters have been added to avoid confusion.

*12. Page 14 Line 23: Add the word 'catchment' so the sentence reads '...a single medium-sized catchment is a challenging...'*

Done

*13. Page 15 Figure 6: Add the following column titles: JP1, MPS, PILB. The same for figures 7 and 8. However I think these figures could all be removed from the manuscript and maybe put in supplementary material in order to cut down the number of figures in the manuscript.*

The suggested column titles have been added.

Again, we would prefer to keep these figures in the manuscript as we think they are relevant for a better understanding of the whole study.

*14. Page 18 Figure 9: What is the CTRL referring to? In the caption replace 'the best and worst ensemble members' with 'the tails of the ensemble'*

CTRL is the acronym of the control (i.e. deterministic simulation). It has been added an explanatory sentence in the caption of Figure 9.

*15. Page 20 Line 8: Are the 7735 grid points just within the catchment or is this over a wider area?*

The 7735 radar grid-points correspond to the radar domain shown in Figures 6 to 8. A clarifying sentence has been added in the text (§ 5.1):

"As the forecast probabilities are computed and verified against each pixel within the radar domain shown in Figures 6 to 8, the statistical sample sums up to 54145 members (7735 radar grid-points times 7 ensemble experiments)."

*16. Page 21 Figure 11: Could a title and units be added to the legends*

Done

[Figure]

**Figure 11: ROC curves of the MPS and PILB ensemble strategies. The embedded figures display the sharpness diagrams containing the number of forecasts used in each probability bin and the total number of observations considered.**

*17. Page 24 Figure 14, 15, 16: I find it hard to see the grey boxes, could these be made a bit darker and maybe thicker so that they stand out more?*

Done

[Figure]

**Figure 14: Peak flow analysis at stations n°2 a) and n°5 b) for 20130305_2d. X-axis shows the delay from the observed peak time, y-axis shows the deviation from the observed peak discharge. The triangles shows the deviation of the simulations with ensemble members forcing (grey for MPS, light blue for PILB), the shapes with black contour shows the deviation of the median of the HEPS simulations with ensemble members forcing, the pink circle shows the deviation of the simulation with JP1 forcing, the green circle the deviation of the simulation with PLU forcing and the dark blue square the deviation of the simulation with deterministic WRF forcing. Alert 1 (yellow dashed line) is the warning threshold, the black star is the observation used as normalized reference.**

[Figure]

**Figure 15: Peak flow analysis at stations n°2 a) and n°5 b) for 20131117_2d. See Figure for the details of the legend.**

[Figure]

a)                                                                                                    b)

**Figure 16: Peak flow analysis at stations n°2 a) and n°5 b) for 20141129_2d. See Figure  for the details of the legend.**

*18. Page 26 Line 18: What is the warning threshold that is used?*

The warning threshold that is used is the the first level alert from the flood warning center in France (SCHAPI). It was already mentioned in §5.2 but has been added in §5.3 for clarity.

*19. Page 27 Line 6: Replace 'excepted' with 'except'*

Done

*20. Page 27 Figure 17, 18, 19: I'm unclear what the two separate graphs in each figure show, could the authors improve the titles and/or captions?*

Done

> **Figure 17: False alarm ratio (FAR) scores at the five gauging stations for the 7 simulations. Statistical indices have been computed by using the observed discharge. Experiments are labelled as WRF: simulated discharge with deterministic WRF forcing, PLU: simulated discharge with PLU forcing, JP1: simulated discharge with JP1 forcing, MPS and PILB: ensemble strategies. The boxplot presents five sample statistics: the minimum, the lower quartile, the median, the upper quartile and the maximum.**

*21. Page 28 Line 5: Define QDF if not already defined*

Quantitative Discharge Forecast (QDF) was defined in the introduction but the acronym has been deleted for clarity reasons.

*22. Page 29 Line 2: Replace 'excepted' with 'except', also occurs on page 30 line 24*

Done

*23. Page 32 Line 12: Could the authors provide more discussion about why there was little difference between the two ensemble strategies? Why were these two different strategies chosen, what differences may have been expected and why do they think these differences weren't observed?*

All these reviewer's concerns have been addressed in the second point of the response letter. We copied the answer below for ease of reading.

When forecasting deep moist convection and heavy rainfall with high-resolution numerical weather prediction models, the outputs are mainly impacted by two sources of errors. One source is the inaccuracies present in the exact representation of the initial and lateral boundary conditions (IC/LBCs). The other source is due to imperfect representation of physical processes via parameterizations. Nowadays, atmospheric ensembles are built to cope with both kinds of distinct uncertainties by perturbing the IC/LBCs or by considering multiple combinations of well-tested numerical schemes. The most appropriate methods for generating Hydrological Ensemble Predictions Systems (HEPS) is a subject under continuous investigation and more methods could come in the future. Here we followed the state-of-the-art approach used in many other hydro-meteorological studies.

Even if PILB and MPS ensembles address different kinds of uncertainties, these sources of error would be expected to have a comparable impact on the skill of quantitative precipitation forecasts (QPFs) if the EPS is properly designed. This seems to be the case of our configuration. Comments on the specific purpose and value of the PILB and MPS ensemble strategies and on the method to avoid under-dispersive behaviour of PILB, have been added in sections 1 and 4.